# Finite-Sample Analysis of Contractive Stochastic Approximation Using Smooth Convex Envelopes

**Zaiwei Chen**
Georgia Institute of Technology
zchen458@gatech.edu

**Siva Theja Maguluri**
Georgia Institute of Technology
siva.theja@gatech.edu

**Sanjay Shakkottai**
The University of Texas at Austin
sanjay.shakkottai@utexas.edu

**Karthikeyan Shanmugam**
IBM Research NY
Karthikeyan.Shanmugam2@ibm.com

## Abstract

Stochastic Approximation (SA) is a popular approach for solving fixed-point equations where the information is corrupted by noise. In this paper, we consider an SA involving a contraction mapping with respect to an arbitrary norm, and show its finite-sample error bounds while using different stepsizes. The idea is to construct a smooth Lyapunov function using the generalized Moreau envelope, and show that the iterates of SA have negative drift with respect to that Lyapunov function. Our result is applicable in Reinforcement Learning (RL). In particular, we use it to establish the first-known convergence rate of the V-trace algorithm for off-policy TD-learning [18]. Importantly, our construction results in only a logarithmic dependence of the convergence bound on the size of the state-space.

## 1 Introduction

Reinforcement Learning (RL) captures an important facet of machine learning going beyond prediction and regression: sequential decision making, and has had great impact in various problems of practical interest [37, 29, 35]. At the heart of RL is the problem of iteratively solving the Bellman's equation using noisy samples, i.e. solving a fixed-point equation of the form $\mathcal{H}(x) = x$. Here, $\mathcal{H}$ is a contractive operator with respect to a suitable norm, where we only have access to samples from noisy versions of the operator. Such fixed-point equations, more broadly, are solved through the framework of Stochastic Approximation (SA) algorithms [33], with several RL algorithms such as $Q$-learning and TD-learning being examples there-of. This paper focuses on understanding the evolution of such a noisy fixed-point iteration through the lens of SA, and providing finite-sample convergence results.

More formally, the SA algorithm for solving the fixed-point equation $\mathcal{H}(x) = x$ is of the form $x_{k+1} = x_k + \epsilon_k (\mathcal{H}(x_k) - x_k + w_k)$, where $\{\epsilon_k\}$ is the stepsize sequence, and $\{w_k\}$ is the noise sequence. To derive finite-sample bounds, three conditions are pertinent: (a) The norm in which the operator $\mathcal{H}$ contracts, (b) The mean zero noise when conditioned on the past, and (c) The nature of the bound on the conditional second moment of the noise.

In prior literature, if the conditional second moment of the noise $\{w_k\}$ is uniformly bounded by a constant, then the norm with respect to which $\mathcal{H}$ being a contraction becomes irrelevant, and it is possible to derive finite-sample convergence guarantees [3, 4, 19, 17]. When the second moment of the noise is not uniformly bounded, then finite-sample bounds can be derived in the case where the norm for contraction of $\mathcal{H}$ is the Euclidean norm [5, 11]. However, in many RL problems, the contraction of $\mathcal{H}$ occurs with respect to a different norm (e.g. the $\ell_\infty$-norm [47] or a weighted variant

[44]). Further, conditioned on the past, the second moment of the norm of the noise scales affinely with the current iterate (again w.r.t. an arbitrary norm), and in general, no uniform bound exists.

An important practical application of this setting with $\ell_\infty$-norm contraction and unbounded noise is the well-known V-trace algorithm for solving the policy evaluation problem using off-policy TD-learning [39]. Its variants form the basis of today's distributed RL platforms like IMPALA [18] and TorchBeast [25] for multi-agent training. It has been used at scale in the recent Deepmind City Navigation Project "Street Learn" [29]. Therefore, deriving finite-sample convergence results for SA under contraction of $\mathcal{H}$ with respect to general norms, and handling unbounded noise are of fundamental interest. In this paper, we answer the following general question in the affirmative:

*Can we provide finite-sample convergence guarantees for the SA algorithm when the norm of contraction of $\mathcal{H}$ is arbitrary, and the second moment of the noise conditioned on the past scales affinely with respect to the squared-norm of the current iterate?*

To the best of our knowledge, except under special conditions on the norm for contraction of $\mathcal{H}$ and/or strong assumptions on the noise, such finite-sample error bounds have not been established. The following table summarizes the results for related works. In Table 1, $d$-dependence refers to the dependence on the dimension $d$ of the iterate $x_k$. To clarify, in the corresponding $d$-dependence of this work for general contractive SA, we write $\log(d)$ ($\|\cdot\|_\infty$) to indicate that the dimension dependence is $\log(d)$ when the norm of contraction is the $\ell_\infty$-norm $\|\cdot\|_\infty$.

Table 1: Comparison to existing bounds

| Topic | Contraction | Noise | Step size | Rate | $d$ dependence |
|---|---|---|---|---|---|
| $Q$-learning [3, 4] | $\|\cdot\|_\infty$ | Bounded | Constant | Geometric | $d^2$ |
| $Q$-learning [46] | $\|\cdot\|_\infty$ | Bounded | $\frac{1}{1+(1-\gamma)k}$ | $O(1/k)$ | $\log(d)$ |
| $Q$-learning [46] | $\|\cdot\|_\infty$ | Bounded | $\frac{1}{k^\xi}$ | $O(1/k^\xi)$ | $\log(d)$ |
| SGD [11] | $\|\cdot\|_2$ | Affine | Constant | Geometric | Independent |
| SGD [11] | $\|\cdot\|_2$ | Affine | $\frac{\beta}{\gamma+k}$ | $O(1/k)$ | Independent |
| $Q$-learning [this work] | $\|\cdot\|_\infty$ | Bounded | Constant | Geometric | $\log(d)$ |
| V-Trace [this work] | $\|\cdot\|_\infty$ | Affine | $\frac{\epsilon}{k+K}$ | $O(1/k)$ | $\log(d)$ |
| Contractive SA [this work] | Arbitrary norm | Affine | Constant & Diminishing | Corollary 2.1 Corollary 2.2 | $\log(d)$ ($\|\cdot\|_\infty$) |

The **main contributions** of this paper are as follows.

**1. Finite-Sample Convergence Guarantees for SA.** We present a novel approach for deriving finite-sample error bounds of the SA algorithm under a general norm contraction. The key idea is to study the drift of a carefully constructed potential/Lyapunov function. We obtain such a potential function by smoothing the norm-squared function using a generalized Moreau envelope. We then study the error bound under either constant or diminishing stepsizes. Specifically, we show that the iterates converge to a ball with radius proportional to the stepsize when using constant stepsize, and converge with rate roughly $O(1/k)$ when using properly chosen diminishing stepsizes.

**2. Performance of the V-trace Algorithm.** To demonstrate the effectiveness of the theoretical result in an entirely novel setting in RL, we consider the V-trace algorithm for solving the policy evaluation problem using off-policy sampling [18]. Interestingly in this case, it is not clear if the iterates of the V-trace algorithm are uniformly bounded by a constant (e.g. as in $Q$-learning [21]). Therefore, existing techniques are not applicable. Using our approach, we establish the first known finite-sample error bounds, and show that the convergence rate is logarithmic in the state-space dimension. In our result, the logarithmic dimension dependence relies on the general form of the Moreau envelope obtained by the infimal convolution with a suitable smooth squared-norm. The freedom in selecting such norm allows us to obtain the logarithmic dependence. Moreover, our approach also recovers

the existing state-of-the-art error bounds [46] for $Q$-learning including logarithmic dependence on dimension (Appendix C), which may be of independent interest.

## 1.1 Summary of our techniques

We now give a more detailed description of the techniques we used. To provide intuition, assume for now that the norm $\|\cdot\|_c$ with respect to which $\mathcal{H}$ being a contraction is the $\ell_p$-norm for $p \in [2, \infty)$, i.e., $\|\mathcal{H}(x) - \mathcal{H}(y)\|_p \leq \gamma\|x - y\|_p$ for all $x, y \in \mathbb{R}^d$, where $\gamma \in (0, 1)$ is the contraction factor. Denote the fixed-point of $\mathcal{H}$ by $x^*$. Consider the Ordinary Differential Equation (ODE) associated with this SA: $\dot{x}(t) = \mathcal{H}(x(t)) - x(t)$. It is shown in [9] (Chapter 10) that $W(x) = \|x - x^*\|_p$ satisfies $\frac{d}{dt}W(x(t)) \leq -\alpha W(x(t))$ for some $\alpha > 0$, which implies the solution $x(t)$ of the ODE converges to its equilibrium point $x^*$ geometrically fast. The term $\alpha$ corresponds to a *negative drift*.

In order to obtain finite-sample bounds, in this paper we study the SA directly, and not the ODE. Then, the Lyapunov function $W(x)$ cannot be directly used to analyze the SA algorithm due to the discretization error and stochastic error. However, suppose we can find a function $M(x)$ that gives negative drift, and in addition: (a) $M(x)$ is $L$ – smooth w.r.t. some norm $\|\cdot\|_s$ [1], (b) the noise $\{w_k\}$ is zero mean conditioned on the past, and (c) the conditional second moment of $\|w_k\|_n$ (where $\|\cdot\|_n$ is again some arbitrary norm) can be bounded affinely by the current iterate $\|x_k\|_n^2$. Then, we have a handle to deal with the discretization error and error caused by the noise to obtain:

$$\mathbb{E}[M(x_{k+1} - x^*)] \leq (1 - O(\epsilon_k) + O(\epsilon_k^2))\mathbb{E}[M(x_k - x^*)] + O(\epsilon_k^2), \tag{1}$$

which implies a contraction in $\mathbb{E}[M(x_{k+1} - x^*)]$. Therefore, a finite-sample error bound can be obtained by recursively applying the previous inequality. The key point is that $M(x)$'s smoothness *and its negative drift with respect to the ODE produces a contraction* $(1 - O(\epsilon_k) + O(\epsilon_k^2))$ *for* $\{x_k\}$. Based on the above analysis, we see that the Lyapunov function for the SA in the case of $\ell_p$-norm contraction should be $M(x) = \frac{1}{2}\|x - x^*\|_p^2$, which is known to be $(p - 1)$ – smooth [1].

However, in the case where $\|\cdot\|_c$ is some arbitrary norm, since the function $f(x) = \frac{1}{2}\|x - x^*\|_c^2$ is not necessarily smooth, the key difficulty is to construct a smooth Lyapunov function. An important special case is when $\|\cdot\|_c = \|\cdot\|_\infty$, which is applicable to many RL algorithms. We provide a solution to this where we construct a smoothed convex envelope $M(x)$ called the *Generalized Moreau Envelope* that is smooth w.r.t. some norm $\|\cdot\|_s$, and it is a tight approximation to $f(x)$, i.e. $aM(x) \leq f(x) \leq bM(x)$ for some constants $a, b > 0$. Further, it is a Lyapunov function for the ODE with a negative drift. This essentially lets us prove a convergence result akin to the case when $f(x)$ is smooth.

## 1.2 Related work

Due to the popularity of the SA algorithm (and its variant Stochastic Gradient Descent (SGD) in optimization [31, 26]), it has been studied extensively in the literature. Specifically, suppose that $\{w_k\}$ is a martingale difference sequence with some mild conditions on its variance, and the stepsize decays to zero at an appropriate rate. Then, almost sure convergence of the sequence $\{x_k\}$ has been established in [44, 23] using a supermartingale convergence approach, and in [10, 9] using an ODE approach. Further, when the iterates are uniformly bounded by an absolute constant (with probability 1), or that the operator $\mathcal{H}$ is contractive with respect to the Euclidean norm, *convergence rates and finite-sample bounds* can be derived using the decomposition methods [44] or Lyapunov techniques [5]. In particular, the decomposition technique has been used for $Q$-learning in [3, 4, 46] to derive finite-sample convergence bounds, using the fact the iterates of $Q$-learning are uniformly bounded by a constant [21]. Concentration results for $Q$-learning were also derived in [32, 28]. As for TD-learning and $Q$-learning with linear function approximation, finite-sample guarantees were shown in [13, 7, 38, 12] for a single-agent problem, and in [16] for a multi-agent problem. Concentration results for SA algorithm when starting near an attractor of the underlying ODE were derived in [8, 43]. Variations of temporal difference (TD) methods (such as gradient TD, least squares TD) have been studied and their convergence has been analyzed in some cases in [48, 41, 40].

Moreau envelopes are popular tools for non-smooth optimization [30], where the proximal operator is used to develop algorithms to work with non-smooth parts of the objective [2]. Moreau envelopes have been used as potential functions to analyze convergence rate of subgradient methods to first order stationary points for non-smooth and non-convex stochastic optimization problems in [14]. They use

the Moreau envelope defined with respect to the Euclidean norm, and use this to show convergence by bounding a measure on first order stationarity with the gradient of the Moreau envelope. In contrast, our interest is in understanding contraction with arbitrary norms – this requires us to use a generalized Moreau envelope obtained by infimal convolution with a general smooth function, and show that its a smooth Lyapunov function with respect to the underlying ODE. The flexibility in the selection of this smooth function in our infimal convolution plays a crucial role in improving the dependence on the state-space dimension to logarithmic factors for our applications.

## 2  Stochastic approximation under a contraction operator

### 2.1  Problem setting: stochastic approximation

Let $\mathcal{H} : \mathbb{R}^d \mapsto \mathbb{R}^d$ be a nonlinear mapping. We are interested in solving for $x^* \in \mathbb{R}^d$ in the equation $\mathcal{H}(x) = x$. Suppose we have access to the mapping $\mathcal{H}$ only through a noisy oracle which for any $x$ returns $\mathcal{H}(x) + w$ ($w$ is the noise). Note that $w$ might depend on $x$. In this setting, the following stochastic iterative algorithm is proposed to estimate $x^*$:

$$x_{k+1} = x_k + \epsilon_k \left( \mathcal{H}(x_k) - x_k + w_k \right), \tag{2}$$

where $\{\epsilon_k\}$ is the stepsize sequence [5]. We next state our main assumptions in studying this SA. Let $\mathcal{F}_k = \{x_0, w_0, ..., x_{k-1}, w_{k-1}, x_k\}$, and let $\| \cdot \|_c$ and $\| \cdot \|_n$ be two arbitrary norms in $\mathbb{R}^d$.

**Assumption 2.1.** *The function $\mathcal{H}$ is a pseudo-contraction mapping w.r.t. norm $\| \cdot \|_c$, i.e., there exists $x^* \in \mathbb{R}^d$ and $\gamma \in (0, 1)$ such that $\|\mathcal{H}(x) - x^*\|_c \leq \gamma \|x - x^*\|_c$ for all $x \in \mathbb{R}^d$.*

**Remark 2.1.** *By letting $x = x^*$, we see that Assumption 2.1 implies $\mathcal{H}(x^*) = x^*$. Moreover, it can be easily shown using proof by contradiction that $x^*$ is the unique fixed-point of $\mathcal{H}$. Note that if $\mathcal{H}$ is indeed a contraction mapping, i.e., $\|\mathcal{H}(x) - \mathcal{H}(y)\|_c \leq \gamma \|x - y\|_c$ for all $x, y \in \mathbb{R}^d$, then by Banach fixed-point theorem [15], $\mathcal{H}$ admits a unique fixed-point $\bar{x}$, and Assumption 2.1 holds with $x^* = \bar{x}$. Hence a contraction is automatically a pseudo-contraction [5].*

**Assumption 2.2.** *The noise sequence $\{w_k\}$ satisfies for all $k \geq 0$: (a) $\mathbb{E}[w_k \mid \mathcal{F}_k] = 0$, and (b) $\mathbb{E}[\|w_k\|_n^2 \mid \mathcal{F}_k] \leq A(1 + \|x_k\|_n^2)$ for some constant $A > 0$.*

Suppose the noise $\|w_k\|_n$ have bounded second moment, or the second moment grows *linearly* in terms of the current iterate $\|x_k\|_n^2$ with a small enough scaling parameter, then finite-sample convergence guarantees were derived in the literature [3, 11]. In this work, our noise assumption is more general in that we allow the conditional variance of $\|w_k\|_n$ to grow *affinely* in terms of the current iterate $\|x_k\|_n^2$, and the scaling parameter $A$ can be arbitrary [5]. This generalization is important when applying our results to Reinforcement Learning.

**Assumption 2.3.** *The stepsize sequence $\{\epsilon_k\}$ is positive and non-increasing.*

The asymptotic convergence of $x_k$ under similar assumptions has been established in the literature. In particular, an approach based on studying the ODE $\dot{x}(t) = \mathcal{H}(x(t)) - x(t)$ was used in [10, 9], where it was shown that $x_k$ converges to $x^*$ almost surely under some stability assumptions of the ODE. The focus of this paper is to establish the finite-sample mean square error bounds for SA algorithm (2). We do this by studying the drift of a smooth potential/Lyapunov function [38, 12]. While we do not explicitly use the ODE approach, the potential function we are going to contruct in the next subsection is inspired by the Lyapunov function used to study the ODE.

### 2.2  The generalized Moreau envelope as a smooth Lyapunov function

Recall from Eq. (1) that with respect to the iterates $\{x_k\}$ of the SA , an ideal Lyapunov function $M(x)$ acts as a potential function that contracts. In this subsection, we first construct a Lyapunov function that is smooth through the generalized Moreau envelope. Smoothness and an approximation property of the Lyapunov function we specify here are used in the next subsection to show the contraction property we desire.

To construct such a Lyapunov function, the following definitions are needed. In this paper, $\langle x, y \rangle = x^\top y$ represents the standard dot product, while the norm $\| \cdot \|$ in the following definition can be any arbitrary norm instead of just being the Euclidean norm $\|x\|_2 = \langle x, x \rangle^{1/2}$.

**Definition 2.1.** *Let $g : \mathbb{R}^d \to \mathbb{R}$ be a convex, differentiable function. Then $g$ is said to be $L$ – smooth w.r.t. the norm $\| \cdot \|$ if and only if $g(y) \leq g(x) + \langle \nabla g(x), y - x \rangle + \frac{L}{2} \|x - y\|^2$ for all $x, y \in \mathbb{R}^d$.*

**Definition 2.2** (Generalized Moreau Envelope [22, 2]). *Let $h_1 : \mathbb{R}^d \mapsto \mathbb{R}$ be a closed and convex function, and let $h_2 : \mathbb{R}^d \mapsto \mathbb{R}$ be a convex and $L$ – smooth function. For any $\mu > 0$, the generalized Moreau envelope of $h_1$ w.r.t. $h_2$ is defined by $M_{h_1}^{\mu, h_2}(x) = \inf_{u \in \mathbb{R}^d} \{h_1(u) + \frac{1}{\mu} h_2(x - u)\}$.*

As an aside, we note that for any two functions $h_1, h_2 : \mathbb{R}^d \mapsto \mathbb{R}$, the function defined by $(h_1 \square h_2)(x) := \inf_{u \in \mathbb{R}^d} \{h_1(u) + h_2(x - u)\}$ is called the infimal convolution of $h_1$ and $h_2$. Therefore, the generalized Moreau envelope in Definition 2.2 can be written as $M_{h_1}^{\mu, h_2}(x) = (h_1 \square \frac{h_2}{\mu})(x)$.

Let $f(x) := \frac{1}{2} \|x\|_c^2$, where $\| \cdot \|_c$ is given in Assumption 2.1. Let $\| \cdot \|_s$ be an arbitrary norm in $\mathbb{R}^d$ such that $g(x) := \frac{1}{2} \|x\|_s^2$ is $L$ – smooth w.r.t. the same norm $\| \cdot \|_s$ in its definition. For example, $\| \cdot \|_s$ can be the $\ell_p$-norm for any $p \in [2, \infty)$ (Example 5.11 [1]). Due to the norm equivalence in $\mathbb{R}^d$ [27], there exist $\ell_{cs}, \ell_{ns} \in (0, 1]$ and $u_{cs}, u_{ns} \in [1, \infty)$ that depend only on the dimenson $d$ and universal constants, such that $\ell_{cs} \| \cdot \|_c \leq \| \cdot \|_s \leq u_{cs} \| \cdot \|_c$ and $\ell_{ns} \| \cdot \|_n \leq \| \cdot \|_s \leq u_{ns} \| \cdot \|_n$.

**Intuition:** With a suitable choice of $\mu$, we will use the Moreau envelope of $f(x)$ with respect to $g(x)$, i.e., $M_f^{\mu, g}(x) = \min_{u \in \mathbb{R}^d} \{f(u) + g(x - u)/\mu\}$ as our Lyapunov function to analyze the behavior of Algorithm (2), where the attainment of the minimum can be justified by Theorem 2.14 of [1]. Intuitively, note that the contraction of $\mathcal{H}$ is w.r.t. $\| \cdot \|_c$, hence the Lyapunov function should be defined in terms of $f(x)$. However, since the function $f(x)$ itself may not be well-behaved (e.g. smooth), we use $g(x)$ as a smoothing function to modify $f(x)$ to obtain $M_f^{\mu, g}(x)$. In order for $M_f^{\mu, g}(x)$ to be a valid Lyapunov function, we need to establish the following two properties: (a) $M_f^{\mu, g}(x)$ should be a smooth function for us to handle the discretization error and the stochastic error in Algorithm (2), and (b) $M_f^{\mu, g}(x)$ should be close to the original function $f(x)$ so that we can use the contraction of $\mathcal{H}$ w.r.t. $\| \cdot \|_c$ to establish the overall contraction of the iterates $\{x_k\}$ w.r.t. $M_f^{\mu, g}(x)$. The following Lemma provides us the desired properties. See Appendix A.1 for its proof.

**Lemma 2.1** (Smoothness and Approximation of the Envelope). *The generalized Moreau envelope $M_f^{\mu, g}(x)$ has the following properties: (a) $M_f^{\mu, g}$ is convex and $L/\mu$ – smooth w.r.t. $\| \cdot \|_s$, (b) we have $(1 + \mu/u_{cs}^2) M_f^{\mu, g}(x) \leq f(x) \leq (1 + \mu/\ell_{cs}^2) M_f^{\mu, g}(x)$ for all $x \in \mathbb{R}^d$, and (c) there exists a norm, denoted by $\| \cdot \|_M$, such that $M_f^{\mu, g}(x) = \frac{1}{2} \|x\|_M^2$ for all $x \in \mathbb{R}^d$.*

Lemma 2.1 (a) is restated from [1], and we include it here for completeness. This, together with Lemma 2.1 (b) implies that $M_f^{\mu, g}(x)$ is a smooth approximation of the function $f(x)$. Lemma 2.1 (c) indicates that $M_f^{\mu, g}(x)$ is in fact a scaled squared norm, and we see from Lemma 2.1 (b) that $\| \cdot \|_M$ gets closer to $\| \cdot \|_c$ when $\mu$ is small.

### 2.3   Recursive contractive bounds for the generalized Moreau envelope

In this subsection, using smoothness of $M_f^{\mu, g}(x)$ and the fact that $M_f^{\mu, g}(x)$ is an approximation to the function $f(x)$ (both properties derived in Lemma 2.1), we derive in the following proposition the desired recursive contraction of $M_f^{\mu, g}(x_k - x^*)$, whose proof is presented in Appendix A.2. To present the coming proposition, we need to define a few more constants. Let

$$\alpha_1 = \frac{1 + \mu/\ell_{cs}^2}{1 + \mu/u_{cs}^2}, \ \alpha_2 = 1 - \gamma \alpha_1^{1/2}, \ \alpha_3 = \frac{4 u_{cs}^2 u_{ns}^2 (A + 2) L (\ell_{cs}^2 + \mu)}{\mu \ell_{cs}^2 \ell_{ns}^2}, \text{ and } \alpha_4 = \frac{\alpha_3 A}{2(A + 2)}.$$

The constant $\mu$ is chosen such that $\alpha_2 > 0$, which is always possible since $\gamma \in (0, 1)$.

**Proposition 2.1.** *The following inequality holds for all $k \geq 0$:*

$$\mathbb{E}[M_f^{\mu, g}(x_{k+1} - x^*) \mid \mathcal{F}_k] \leq (1 - 2\alpha_2 \epsilon_k + \alpha_3 \epsilon_k^2) M_f^{\mu, g}(x_k - x^*) + \frac{\alpha_4 (1 + 2\|x^*\|_c^2)}{2(1 + \mu/\ell_{cs}^2)} \epsilon_k^2. \quad (3)$$

From Eq. (3), we see that $\alpha_2$ represents the real contraction effect of the algorithm, and it should be positive, which leads to our feasible range of $\mu$. On the r.h.s. of Eq. (3), the first term represents the *overall contraction* property that results from a combination of the contraction in the drift term

that counteracts an expansion resulting from the discretization error and the noise second moment that scales affinely in $\|x_k\|_n^2$. The second term is a consequence of discretization and the noise $\{w_k\}$. This is the key step in our proof compared to [3, 4, 32] as *we do not decompose the analysis into one for the contraction terms and another for the noise terms.*

## 2.4 Putting together: finite-sample bounds for SA

From Proposition 2.1, the finite-sample error bound of Algorithm (2) can be established by repeatedly using Eq. (3), which leads to our main result in the following. See Appendix A.3 for its proof.

**Theorem 2.1.** *Consider iterates $\{x_k\}$ of Algorithm (2). Suppose Assumptions 2.1 – 2.3 are satisfied, and $\epsilon_0 \leq \alpha_2/\alpha_3$. Then we have for all $k \geq 0$:*

$$\mathbb{E}\left[\|x_k - x^*\|_c^2\right] \leq \alpha_1 \|x_0 - x^*\|_c^2 \prod_{j=0}^{k-1}(1 - \alpha_2\epsilon_j) + \alpha_4(1 + 2\|x^*\|_c^2)\sum_{i=0}^{k-1}\epsilon_i^2 \prod_{j=i+1}^{k-1}(1 - \alpha_2\epsilon_j). \quad (4)$$

In Eq. (4), the first term represents how fast the initial condition is forgotten, hence it is proportional to the error in our initial guess $\|x_0 - x^*\|_c^2$. The second term represents the impact of the variance in our estimate. Note that the condition $\epsilon_0 \leq \alpha_2/\alpha_3$ is made only for ease of exposition. If it is not true, as long as $\lim_{k\to\infty}\epsilon_k < \alpha_2/\alpha_3$, we can let $k' := \min\{k \geq 0 : \epsilon_k \leq \alpha_2/\alpha_3\}$, and then recursively apply Eq. (3) starting from the $k'$-th iteration. For $k = 0, 1, ..., k'$, it can be easily shown using Eq. (3) that $\mathbb{E}[\|x_k - x^*\|_c^2]$ is bounded.

*Theorem 2.1 is our key contribution in that it holds in the case when: (a) the contraction of $\mathcal{H}$ can be w.r.t. any general norms, and (b) the conditional second moment of the noise is not bounded by a constant but in fact scales affinely in the current iterates (see Assumption 2.2). As far as we are aware, Theorem 2.1 establishes the first-known finite-sample convergence bounds in these general settings.*

## 2.5 Results with various stepsize regimes and dimension dependence

Upon obtaining a finite-sample error bound in its general form in Theorem 2.1, we next consider two common choices of stepsizes, and see what does Eq. (4) give us. We first consider using constant stepsize (i.e., $\epsilon_k \equiv \epsilon \leq \alpha_2/\alpha_3$) in the following result, whose proof is presented in Appendix A.4.

**Corollary 2.1.** $\mathbb{E}\left[\|x_k - x^*\|_c^2\right] \leq \alpha_1 \|x_0 - x^*\|_c^2(1 - \alpha_2\epsilon)^k + (1 + 2\|x^*\|_c^2)\frac{\alpha_4\epsilon}{\alpha_2}$ *for all $k \geq 0$.*

From Corollary 2.1, we see that in expectation, the iterates converge exponentially fast in the mean square sense, to a ball with radius proportional to the stepsize $\epsilon$, centered at the fixed-point $x^*$. With smaller stepsize, at the end the estimate $x_k$ of $x^*$ is more accurate, but the rate of convergence is slower since the geometric ratio $(1 - \alpha_2\epsilon)$ is larger.

We next consider using diminishing stepsizes of the form $\epsilon_k = \epsilon/(k + K)^\xi$, where $\epsilon > 0$, $\xi \in (0, 1]$, $K = \max(1, \epsilon\alpha_3/\alpha_2)$ when $\xi = 1$, and $K = \max(1, (\epsilon\alpha_3/\alpha_2)^{1/\xi}, [2\xi/(\alpha_2\epsilon)]^{1/(1-\xi)})$ when $\xi \in (0, 1)$. The main reason for introducing $K$ here is to make sure that $\epsilon_0 \leq \alpha_2/\alpha_3$. We have the following result, whose proof is presented in Appendix A.5.

**Corollary 2.2.** *Suppose $\epsilon_k$ is of the form given above, then we have*

*(a)* $\mathbb{E}[\|x_k - x^*\|_c^2] \leq \alpha_1 \|x_0 - x^*\|_c^2 \left(\frac{K}{k+K}\right)^{\alpha_2\epsilon} + \frac{4\epsilon^2\alpha_4}{1-\alpha_2\epsilon}\frac{1+2\|x^*\|_c^2}{(k+K)^{\alpha_2\epsilon}}$ *when $\xi = 1$, and $\epsilon < 1/\alpha_2$.*

*(b)* $\mathbb{E}[\|x_k - x^*\|_c^2] \leq \alpha_1 \|x_0 - x^*\|_c^2 \frac{K}{k+K} + \frac{4\alpha_4}{\alpha_2^2}\frac{(1+2\|x^*\|_c^2)\log(k+K)}{k+K}$ *when $\xi = 1$, and $\epsilon = 1/\alpha_2$.*

*(c)* $\mathbb{E}[\|x_k - x^*\|_c^2] \leq \alpha_1 \|x_0 - x^*\|_c^2 \left(\frac{K}{k+K}\right)^{\alpha_2\epsilon} + \frac{4\epsilon^2\alpha_4}{\alpha_2\epsilon-1}\frac{1+2\|x^*\|_c^2}{k+K}$ *when $\xi = 1$, and $\epsilon > 1/\alpha_2$.*

*(d)* $\mathbb{E}[\|x_k - x^*\|_c^2] \leq \alpha_1 \|x_0 - x^*\|_c^2 \exp\left\{-\frac{\alpha_2\epsilon}{1-\xi}\left[(k+K)^{1-\xi} - K^{1-\xi}\right]\right\} + \frac{2\epsilon\alpha_4}{\alpha_2}\frac{1+2\|x^*\|_c^2}{(k+K)^\xi}$ *when $\xi \in (0, 1)$, and $\epsilon > 0$.*

According to Corollary 2.2, when the stepsizes are chosen as $\epsilon_k = \epsilon/(k + K)$, the constant $\epsilon$ is important in determining the convergence rate, and the best convergence rate of $O(1/k)$ is attained when $\epsilon > 1/\alpha_2$. This is because the constant $\alpha_2$ (see Eq. (4)) represents the real contraction effect of the algorithm. When $\alpha_2$ is small, we choose large $\epsilon$ to compensate for the slow contraction.

If $\xi \in (0, 1)$, the convergence rate is roughly $O(1/k^\xi)$, which is sub-optimal but more robust, since the rate does not depend on the choice of $\epsilon$. This suggests the following rule of thumb in tuning the

stepsizes. If we know the contraction factor $\gamma$, we know $\alpha_2$ given in Proposition 2.1 (since we pick $g(x)$ and $\mu$ ). Thus by choosing $\epsilon_k = \epsilon/(k+K)$ with $\epsilon > 1/\alpha_2$, we achieve the optimal convergence rate. When our estimate of $\gamma$ is poor, to avoid being in case (a) of Corollary 2.2, it is better to use $\epsilon_k = \epsilon/(k+K)^\xi$ as the stepsize, thereby trading-off between convergence rate and robustness.

**Connection to SGD:** Although Theorem 2.1 (and Corollaries 2.1, 2.2) are derived for SA algorithms involving a contraction operator, they also recover finite-sample bounds for SGD with a smooth and strongly convex objective, whose convergence rate is $O(1/k)$. To see this, let $F(x)$ be a differentiable objective function which is smooth and strongly convex with parameters $C$ and $c$. Define the operator $\mathcal{H}$ by $\mathcal{H}(x) = -\eta \nabla F(x) + x$, where $\eta > 0$. Then Algorithm (2) becomes $x_{k+1} = x_k + \epsilon_k(-\eta \nabla F(x_k) + w_k)$, which is the SGD algorithm for minimizing $F(x)$ [26, 31]. Further, it is known that $\mathcal{H}$ is a Lipschitz operator w.r.t. the Euclidean norm, with Lipschitz constant $L_{SGD} = \max(|1 - \eta c|, |1 - \eta C|)$ [34]. When $\eta \in (0, 2/C)$, we have $L_{SGD} < 1$, and hence the operator $\mathcal{H}$ is a contraction with respect to the Euclidean norm.

**Logarithmic Dependence on Dimension:** Switching focus, we next show in the following Corollary that with suitable choices of $g(x)$ (i.e. $\|\cdot\|_s$) and $\mu$, our approach naturally results in only logarithmic dependence on the dimension $d$ in the case where both $\|\cdot\|_c$ and $\|\cdot\|_n$ are the $\ell_\infty$-norm. The case of $\ell_\infty$-norm contraction is of special interest due to its applications in RL. We will use this result in Section 3, to analyze the V-trace algorithm for off-policy TD-learning and in Appendix C for the $Q$-learning algorithm.

**Corollary 2.3.** *Consider the case where $\|\cdot\|_c = \|\cdot\|_n = \|\cdot\|_\infty$. Let $g(x) = \frac{1}{2}\|x\|_p^2$ with $p = 4\log(d)$ and let $\mu = (1/2 + 1/(2\gamma))^2 - 1$. Then we have $\alpha_1 \leq \frac{3}{2}$, $\alpha_2 \geq \frac{1}{2}(1-\gamma)$, $\alpha_3 \leq \frac{32e(A+2)\log(d)}{1-\gamma}$, and $\alpha_4 \leq \frac{16eA\log(d)}{1-\gamma}$. (See Appendix A.6 for the proof)*

**Order-Wise Tightness:** In general, we cannot hope to improve the convergence rate beyond $O(1/k)$ or the dimension dependence beyond $\log(d)$. To see this, consider the trivial case where $\mathcal{H}(x)$ is identically zero, and $\{w_k\}$ is an i.i.d. sequence of standard normal random vectors. Algorithm (2) becomes $x_{k+1} = x_k + \epsilon_k(-x_k + w_k)$, which can be viewed as an SA algorithm for solving the trivial equation $x = 0$, or an SGD algorithm for minimizing a quadratic objective $F(x) = \frac{1}{2}\|x\|_2^2$. When $\epsilon_k = \frac{1}{k+1}$, the iterates $x_k$ are just the running averages of $\{w_k\}$, i.e., $x_k = \frac{1}{k}\sum_{i=0}^{k-1} w_i$ for all $k \geq 1$, which implies $\sqrt{k}x_k \sim \mathcal{N}(0, I)$. Since it is well-known that $\mathbb{E}[\|X\|_\infty^2] \sim O(\log(d))$ for a standard normal random vector $X$ [45], we have $\mathbb{E}[\|x_k\|_\infty^2] = O(\frac{\log(d)}{k})$. Thus in this setting, our resulting finite-sample bounds under $\ell_\infty$-norm contraction are order-wise tight both in terms of the convergence rate and the dimensional dependence.

**In summary**, we have (a) stated and proved a finite-sample error bound for Algorithm (2) in its general form (Theorem 2.1), (b) studied its behavior under different choices of stepsizes (Corollaries 2.1 and 2.2), and (c) elaborated how to choose the function $g(x)$ and the parameter $\mu$ used in the generalized Moreau envelope to optimize the constants in the bound (4) (Corollary 2.3). In the next section, we present how the convergence results in this section apply in the context of RL.

## 3 Applications in Reinforcement Learning

### 3.1 Overview and notation

We study the infinite-horizon discounted (with discount factor $\beta \in (0, 1)$) Markov Decision Process (MDP) $\mathcal{M} = \{\mathcal{S}, \mathcal{A}, \mathcal{P}, \mathcal{R}\}$. Here, $\mathcal{S}$ is the finite state-space ($|\mathcal{S}| = n$), $\mathcal{A}$ is the finite action-space ($|\mathcal{A}| = m$), $\mathcal{P} = \{P_a \in \mathbb{R}^{n \times n} \mid a \in \mathcal{A}\}$ is the set of unknown action dependent transition probability matrices, and $\mathcal{R} : \mathcal{S} \times \mathcal{A} \mapsto \mathbb{R}$ is the reward function. Since we work with finite state-action spaces, we can without loss of generality assume $\mathcal{R}(s, a) \in [0, 1]$ for all $(s, a)$. See [39] for more details about MDP. The goal in RL is to find a policy $\pi^*$ (aka the optimal policy) that maximizes the expected total reward. Specifically, the value of a policy $\pi$ at state $s$ is defined by $V_\pi(s) = \mathbb{E}[\sum_{k=0}^\infty \beta^k r_k \mid S_0 = s]$, where $r_k := \mathcal{R}(S_k, A_k)$, and $A_k$ is executed according the policy $\pi$. We want to find $\pi^*$ so that $V_{\pi^*}(s) \geq V_\pi(s)$ for all $\pi$ and $s$.

The convergence of many classical algorithms for solving the RL problem such as TD-learning (e.g. TD(0), TD($n$), and TD($\lambda$)) and $Q$-learning relies on the stochastic approximation under contraction assumption [5]. Therefore, our result is a broad tool to establish the finite-sample error bounds of

various RL algorithms. We next present a case study on the V-trace algorithm [18] for solving the policy evaluation problem using off-policy sampling. Our result can also be used to recover the existing state-of-the-art finite-sample bounds of $Q$-learning [3, 46] (See Appendix C).

## 3.2 The V-trace algorithm for off-policy Reinforcement Learning

One popular approach for finding $\pi^*$ is through the following iteration: with some initialization policy $\pi_0$, for any $k \geq 0$, first estimate the value function $V_{\pi_k}$, then update the policy to $\pi_{k+1}$ with some strategy (e.g. policy gradient), and repeat this process until $\pi_k$ closely represents $\pi^*$. An important intermediate step here is to estimate $V_\pi$ for a given policy $\pi$, which is called the policy evaluation problem [39]. Since we do not have access to the system parameters $\mathcal{P}$, a popular method for solving the policy evaluation problem is the TD-learning method [42], where one tries to estimate $V_\pi$ using the samples collected from the system.

In off-policy TD-learning algorithms [39], one uses trajectories generated by a *behavior policy* $\pi' \neq \pi$ to learn the value function of the *target policy* $\pi$. Off-policy methods are used for three important reasons in the TD-setting: (a) It is typically necessary to have an exploration component in the behavior policy $\pi'$ which makes it different from the target policy $\pi$. (b) It is used in multi-agent training where various agents collect rewards using a behavior policy that is lagging with respect to the target policy in an actor-critic framework [18]. (c) One set of samples can be used more than once to evaluate different target policies, which can leverage acquired data in the past.

Off-policy TD-learning is implemented through importance sampling to obtain an unbiased estimate of $V_\pi$. However, the variance in the estimate can blow up since the importance sampling ratio can be very large [20]. Thus, a well-known and fundamental difficulty in off-policy TD-learning with importance sampling is that of balancing the bias-variance trade-off.

Recently, [18] proposed an off-policy TD-learning algorithm called the V-trace, where they introduced two truncation levels in the importance sampling weights. Their construction (through two separate clippers) crucially allows the algorithm to control the bias in the limit (through one clipper), while the other clipper mainly controls the variance in the estimate. The V-trace algorithm has had a huge practical impact: it has been implemented in distributed RL architectures and platforms like IMPALA (a Tensorflow implementation) [18] and TorchBeast (a PyTorch implementation) [25] for multi-agent training besides being used at scale in a recent Deepmind City Navigation Project "Street Learn" [29]. Given its impact, a theoretical understanding of the effects of the truncation levels on convergence rate is important for determining how to tune them to improve the performance of V-trace.

In this paper we consider a synchronous version of the V-trace algorithm. Let $\pi'$ be a behavior policy used to collect samples, and let $\pi$ be the target policy whose value function is to be estimated. We first initialize $V_0 \in \mathbb{R}^n$. Given a fixed horizon $T > 0$, in each iteration $k \geq 0$, for each state $s \in \mathcal{S}$, a trajectory $\{S_0, A_0, ..., S_T, A_T\}$ with initial state $S_0 = s$ is generated using the behavior policy $\pi'$. Then, the corresponding entry of the estimate $V_k$ is updated according to

$$V_{k+1}(s) = V_k(s) + \epsilon_k \sum_{t=0}^{T} \beta^t \left( \prod_{j=0}^{t-1} c_j \right) \rho_t \left( r_t + \beta V_k(S_{t+1}) - V_k(S_t) \right), \qquad (5)$$

where $c_t = \min\left(\bar{c}, \frac{\pi(A_t|S_t)}{\pi'(A_t|S_t)}\right)$ and $\rho_t = \min\left(\bar{\rho}, \frac{\pi(A_t|S_t)}{\pi'(A_t|S_t)}\right)$ are truncated importance sampling weights with truncation levels $\bar{\rho} \geq \bar{c}$. Here we use the convention that $c_t = \rho_t = 1$, and $r_t = 0$ whenever $t < 0$. In the special case where the behavior policy $\pi'$ and the target policy $\pi$ coincide, and $\bar{c} \geq 1$, Algorithm (5) boils down to the on-policy multi-step TD-learning update [39]. To simplify the notation, we denote $c_{a,b} = \prod_{t=a}^{b} c_t$ in the following.

The asymptotic convergence of Algorithm (5) when $T = \infty$ has been established in [18] using the convergence results of stochastic approximation under contraction assumptions [5, 24]. The quality of the V-trace limit as a function of $\bar{\rho}$ and $\bar{c}$ has also been discussed in [18]. Specifically, $\bar{\rho}$ alone determines the limiting value function, and $\bar{c}$ mainly controls the variance in the estimates $\{V_k\}$.

**Properties of the V-Trace Algorithm:** Our goal is to understand the convergence rate of Algorithm (5) for any choices of $\bar{\rho}$ and $\bar{c}$, which will determine the bias-vs-convergence-rate trade-off. First of all, when $T < \infty$, similarly as in [18], Algorithm (5) admits the following properties (See Appendix B.1 for the proof):

(a) Algorithm (5) can be rewritten (in the vector form) as $V_{k+1} = V_k + \epsilon_k(\mathcal{H}(V_k) - V_k + w_k)$.

(b) Suppose the behavior policy $\pi'$ satisfies the *coverage assumption*, i.e., $\{a \mid \pi(a|s) > 0\} \subseteq \{a \mid \pi'(a|s) > 0\}$ for all $s \in \mathcal{S}$. Then the mapping $\mathcal{H}$ is a contraction w.r.t. $\|\cdot\|_\infty$, and the contraction factor is $\gamma := 1 - (1 - \beta) \min_{s \in \mathcal{S}} \sum_{t=0}^{T} \beta^t \mathbb{E}_{\pi'}[c_{0,t-1}\rho_t \mid S_0 = s] \leq 1 - (1 - \beta)\zeta < 1$, where $\zeta > 0$ is a constant.

(c) $\mathcal{H}$ has a unique fixed-point $V_{\pi_{\bar\rho}}$, where $\pi_{\bar\rho}(a|s) = \frac{\min(\bar\rho\pi'(a|s),\pi(a|s))}{\sum_{b \in \mathcal{A}} \min(\bar\rho\pi'(b|s),\pi(b|s))}$ for all $(s,a)$. Note that when $\bar\rho \geq \rho_{\max} := \max_{(s,a)} \frac{\pi(a|s)}{\pi'(s|a)}$, we have $\pi_{\bar\rho} = \pi$. Otherwise the policy $\pi_{\bar\rho}$ is in some sense between the behavior policy $\pi'$ and the target policy $\pi$. The difference between $V_{\pi_{\bar\rho}}$ and $V_\pi$ due to the truncation can be controlled as: $\|V_{\pi_{\bar\rho}} - V_\pi\|_\infty \leq \frac{2}{(1-\gamma)^2}\|\pi_{\bar\rho} - \pi\|_\infty$.

(d) The noise sequence $\{w_k\}$ satisfies Assumption 2.2 with $\|\cdot\|_n = \|\cdot\|_\infty$ and $A = \frac{32\bar\rho^2}{(1-\beta\bar c)^2}$ when $\beta\bar c < 1$, $A = 32\bar\rho^2(T+1)^2$ when $\beta\bar c = 1$, and $A = \frac{32\bar\rho^2(\beta\bar c)^{2T+2}}{(\beta\bar c - 1)^2}$ when $\beta\bar c > 1$.

### 3.3  Finite-sample analysis of the V-trace algorithm

The properties of the V-trace algorithm indicate that Assumptions 2.1 and 2.2 are satisfied for Algorithm (5). Let us now use our results in Section 2 to establish a finite-sample error bound of $\{V_k\}$ and study its dependence on the two truncation levels $\bar c$, $\bar\rho$, and the horizon $T$. Observe that $\|\cdot\|_c = \|\cdot\|_n = \|\cdot\|_\infty$ in this problem, Corollary 2.3 is applicable. For ease of exposition, here we only consider the $O(1/k)$ stepsizes, and pick the parameters to ensure that we fall in case (c) of Corollary 2.2, which has the best convergence rate. The finite-sample error bound for other cases can be derived similarly. The proof of the following theorem is presented in Appendix B.2.

**Theorem 3.1.** *Consider* $\{V_k\}$ *of Algorithm (5). Suppose that* $\epsilon_k = \frac{\epsilon}{k+K}$ *with* $\epsilon = \frac{4}{1-\gamma}$ *and* $K = \frac{64(A+2)\log|\mathcal{S}|}{(1-\gamma)^3}$. *Then we have for all* $k \geq 0$:

$$\mathbb{E}\left[\|V_k - V_{\pi_{\bar\rho}}\|_\infty^2\right] \leq 1024e^2(\|V_0 - V_{\pi_{\bar\rho}}\|_\infty^2 + 2\|V_{\pi_{\bar\rho}}\|_\infty^2 + 1)\frac{(A+2)\log|\mathcal{S}|}{(1-\gamma)^3}\frac{1}{k+K}. \quad (6)$$

To better understand the how the parameters $\bar c$, $\bar\rho$, and $T$ impact the convergence rate. Suppose we want to find the required number of iterations so that in expectation the distance between $x_k$ and $x^*$ is less than $\delta$, i.e., $k_\delta = \min\{k \geq 0 : \mathbb{E}[\|x_k - x^*\|_\infty^2] \leq \delta\}$. Using Eq. (6) and we have $k_\delta \geq 1024e^2(\|V_0 - V_{\pi_{\bar\rho}}\|_\infty^2 + 2\|V_{\pi_{\bar\rho}}\|_\infty^2 + 1)\frac{(A+2)\log|\mathcal{S}|}{\delta(1-\gamma)^3}$. We first note that the dimension dependence of $k_\delta$ is only $\log|\mathcal{S}|$. The parameters $\bar c$, $\bar\rho$, and $T$ impact the convergence rate through $A$ and $\gamma$. Though $\gamma$ is a decreasing function of $\bar c$, $\bar\rho$, and $T$, it can be upper bounded by $1 - (1-\beta)\zeta$ (see property (b) of the V-trace algorithm). It follows that the term $1/(1-\gamma)^3$ can be bounded above by $1/[(1-\beta)^3\zeta^3]$. Therefore, the main impact comes through the constant $A = A(\bar c, \bar\rho, T)$.

From property (d) of the V-trace algorithm, we see that $A$ is a piecewise function of $\bar c$, $\bar\rho$, and $T$. In all its cases, $\bar\rho$ appears quadratically in $A(\bar c, \bar\rho, T)$. The impact of $\bar c$ and $T$ is more subtle. When $\beta\bar c < 1$, $A(\bar c, \bar\rho, T)$ is independent of the horizon $T$. However, when $\beta\bar c = 1$ or $\beta\bar c > 1$, $A(\bar c, \bar\rho, T)$ increases either linearly or exponentially in terms of $T$, which suggests that $\bar c < 1/\beta$ is a better choice. Such a small $\bar c$ can lead to the contraction factor $\gamma$ being close to unity (See property (b)), which increases the error in Eq. (6). However, since $A$ does not depend on $T$ when $\bar c < 1/\beta$, this drawback can be avoided by increasing the horizon $T$, which decreases the contraction parameter $\gamma$, albeit at the cost of more samples in each iterate. For a given application, based on the above idea, we can numerically optimize the parameters ($\bar c$, $\bar\rho$, and $T$) to trade-off between contraction factor and variance.

Though we have analyzed the convergence rate of $V_k$, the limiting value function $V_{\pi_{\bar\rho}}$ is not the value function of the target policy $\pi$. Note that this bias can be eliminated by choosing $\bar\rho \geq \rho_{\max}$, provided that $\rho_{\max}$ is finite. However, when the number of state-action pairs is infinite, and when we use V-trace algorithm along with function approximation, $\rho_{\max}$ can be infinity. Studying such a scenario is one of our future directions.

**In summary**, we have established the first-known finite-sample error bound of the V-trace algorithm using our general results on SA in Section 2. Moreover, from the resulting bound (6), we analyzed how the parameters of the problem (i.e., the two truncation levels $\bar c$, $\bar\rho$, and the horizon $T$) impact the convergence rate, and provided a rule of thumb in tuning them.

## Broader Impact

This work focuses on theoretical results for stochastic approximation, which are then applied for understanding the properties of Reinforcement Learning algorithms. While RL algorithms have important societal implications (e.g. in autonomous driving, RL algorithms for network control, etc.), and thus understanding their performance is important, we believe that the direct ethical consequences of our work is somewhat limited.

## Acknowledgments and Disclosure of Funding

This work was partially supported by ONR Grant N00014-19-1-2566, NSF Grant CNS-1910112, ARO grant W911NF-17-1-0359, ONR Grant N00014-19-1-2566, NSF Grant CNS-1910112, ARO grant W911NF-17-1-0359, and Raytheon Technologies. Maguluri also acknowledges seed funding from Georgia Institute of Technology.

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
