[Supplementary Material]

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

# Appendices

## A    Proofs of All Technical Results in Section 2

### A.1    Proof of Lemma 2.1

(a) The convexity of $M_f^{\mu,g}(x)$ follows from Theorem 2.19 of [1]. Since $f(x)$ is proper, closed, and convex, $g(x)$ is $L$ – smooth with respect to $\|\cdot\|_s$, we have by [1] Theorem 5.30 (a) that $M_f^{\mu,g}(x) = (f \square \frac{g}{\mu})(x)$ is $\frac{L}{\mu}$ – smooth with respect to $\|\cdot\|_s$.

(b) We first derive the upper bound of $f(x)$. By definition of $M_f^{\mu,g}(x)$, we have

$$
\begin{aligned}
M_f^{\mu,g}(x) &= \min_{u \in \mathbb{R}^d} \left\{ \frac{1}{2}\|u\|_c^2 + \frac{1}{2\mu}\|x-u\|_s^2 \right\} \\
&\geq \min_{u \in \mathbb{R}^d} \left\{ \frac{1}{2}\|u\|_c^2 + \frac{\ell_{cs}^2}{2\mu}\|x-u\|_c^2 \right\} && (\ell_{cs}\|\cdot\|_c \leq \|\cdot\|_s) \\
&\geq \min_{u \in \mathbb{R}^d} \left\{ \frac{1}{2}\|u\|_c^2 + \frac{\ell_{cs}^2}{2\mu}(\|x\|_c - \|u\|_c)^2 \right\} && \text{(Triangle inequality)} \\
&= \min_{y \in \mathbb{R}} \left\{ \frac{1}{2}y^2 + \frac{\ell_{cs}^2}{2\mu}(\|x\|_c - y)^2 \right\} && \text{(change of variable: } y = \|u\|_c^2) \\
&= \min_{y \in \mathbb{R}} \left\{ \left(\frac{1}{2} + \frac{\ell_{cs}^2}{2\mu}\right)y^2 - \frac{\ell_{cs}^2}{\mu}\|x\|_c y + \frac{\ell_{cs}^2}{2\mu}\|x\|_c^2 \right\} \\
&= \frac{1}{2}\|x\|_c^2 \frac{\ell_{cs}^2}{\mu + \ell_{cs}^2} && \text{(minimum of a quadratic function)} \\
&= \frac{\ell_{cs}^2}{\mu + \ell_{cs}^2} f(x).
\end{aligned}
$$

It follows that $f(x) \leq \left(1 + \mu/\ell_{cs}^2\right) M_f^{\mu,g}(x)$ for all $x$. Next we show the lower bound. Similarly, by definition we have

$$
\begin{aligned}
M_f^{\mu,g}(x) &= \min_{u \in \mathbb{R}^d} \left\{ \frac{1}{2}\|u\|_c^2 + \frac{1}{2\mu}\|x-u\|_s^2 \right\} \\
&\leq \min_{\alpha \in (0,1)} \left\{ \frac{1}{2}\|\alpha x\|_c^2 + \frac{1}{2\mu}\|x - \alpha x\|_s^2 \right\} && \text{(restrict } u = \alpha x \text{ for } \alpha \in (0,1)) \\
&\leq \frac{1}{2}\|x\|_c^2 \min_{\alpha \in (0,1)} \left\{ \alpha^2 + \frac{(1-\alpha)^2 u_{cs}^2}{\mu} \right\} && (\|\cdot\|_s \leq u_{cs}\|\cdot\|_c) \\
&= \frac{u_{cs}^2}{u_{cs}^2 + \mu} \frac{1}{2}\|x\|_c^2 && \text{(minimum of the quadratic function)} \\
&= \frac{u_{cs}^2}{u_{cs}^2 + \mu} f(x).
\end{aligned}
$$

It follows that $f(x) \geq \left(1 + \mu/u_{cs}^2\right) M_f^{\mu,g}(x)$ for all $x$.

(c) It is clear from the definition of $M_f^{\mu,g}(x)$ that it is non-negative and is equal to zero if and only if $x = 0$. Now for any $\alpha \in \mathbb{R}$, we have

$$
\begin{aligned}
M_f^{\mu,g}(\alpha x) &= \min_u \left\{ \frac{1}{2}\|u\|_c^2 + \frac{1}{2\mu}\|\alpha x - u\|_s^2 \right\} \\
&= \min_v \left\{ \frac{1}{2}\|\alpha v\|_c^2 + \frac{1}{2\mu}\|\alpha x - \alpha v\|_s^2 \right\} && \text{(change of variable } u = \alpha v) \\
&= |\alpha|^2 M_f^{\mu,g}(x).
\end{aligned}
$$

Thus, $\sqrt{M_f^{\mu,g}(\alpha x)} = |\alpha|\sqrt{M_f^{\mu,g}(x)}$. It remains to show the Triangle inequality. For any $x_1, x_1 \in \mathbb{R}^d$, let $u_1 \in \arg\min_{u \in \mathbb{R}^d}\{\frac{1}{2}\|u\|_c^2 + \frac{1}{2\mu}\|x_1 - u\|_s^2\}$ and $u_2 \in \arg\min_{u \in \mathbb{R}^d}\{\frac{1}{2}\|u\|_c^2 + \frac{1}{2\mu}\|x_2 - u\|_s^2\}$. Then we have

$$
\begin{aligned}
& M_f^{\mu,g}(x_1 + x_2) \\
&= \min_u \left\{ \frac{1}{2}\|u\|_c^2 + \frac{1}{2\mu}\|x_1 + x_2 - u\|_s^2 \right\} \\
&\leq \frac{1}{2}\|u_1 + u_2\|_c^2 + \frac{1}{2\mu}\|x_1 + x_2 - u_1 - u_2\|_s^2 && \text{(choose } u = u_1 + u_2\text{)} \\
&\leq \frac{1}{2}(\|u_1\|_c + \|u_2\|_c)^2 + \frac{1}{2\mu}(\|x_1 - u_1\|_s + \|x_2 - u_2\|_s)^2 \\
&= M_f^{\mu,g}(x_1) + M_f^{\mu,g}(x_2) + \|u_1\|_c\|u_2\|_c + \frac{1}{\mu}\|x_1 - u_1\|_s\|x_2 - u_2\|_s \\
&\leq M_f^{\mu,g}(x_1) + M_f^{\mu,g}(x_2) + 2\sqrt{\frac{1}{2}\|u_1\|_c^2 + \frac{1}{2\mu}\|x_1 - u_1\|_s^2}\sqrt{\frac{1}{2}\|u_2\|_c^2 + \frac{1}{2\mu}\|x_2 - u_2\|_s^2} \\
&= M_f^{\mu,g}(x_1) + M_f^{\mu,g}(x_2) + 2\sqrt{M_f^{\mu,g}(x_1)M_f^{\mu,g}(x_2)}.
\end{aligned}
$$

It follows that $\sqrt{M_f^{\mu,g}(x_1 + x_2)} \leq \sqrt{M_f^{\mu,g}(x_1)} + \sqrt{M_f^{\mu,g}(x_2)}$ for any $x_1, x_2 \in \mathbb{R}^d$. Therefore, we can write $M_f^{\mu,g}(x)$ as $\frac{1}{2}\|x\|_M^2$ for some norm $\|\cdot\|_M$.

## A.2 Proof of Proposition 2.1

Using Lemma 2.1 (a) and Algorithm (2), we have

$$
\begin{aligned}
M_f^{\mu,g}(x_{k+1} - x^*) &\leq M_f^{\mu,g}(x_k - x^*) + \langle \nabla M_f^{\mu,g}(x_k - x^*), x_{k+1} - x_k \rangle + \frac{L}{2\mu}\|x_{k+1} - x_k\|_s^2 \\
&= M_f^{\mu,g}(x_k - x^*) + \epsilon_k \langle \nabla M_f^{\mu,g}(x_k - x^*), \mathcal{H}(x_k) - x_k \rangle \\
&\quad + \epsilon_k \langle \nabla M_f^{\mu,g}(x_k - x^*), w_k \rangle + \frac{L\epsilon_k^2}{2\mu}\|\mathcal{H}(x_k) - x_k + w_k\|_s^2.
\end{aligned}
$$

Taking expectation conditioned on $\mathcal{F}_k$ on both side of the previous inequality then using Assumption 2.2 (a), we have

$$
\mathbb{E}[M_f^{\mu,g}(x_{k+1} - x^*) \mid \mathcal{F}_k] \leq M_f^{\mu,g}(x_k - x^*) + \epsilon_k \underbrace{\langle \nabla M_f^{\mu,g}(x_k - x^*), \mathcal{H}(x_k) - x_k \rangle}_{E_1}
$$

$$
+ \frac{L\epsilon_k^2}{2\mu} \underbrace{\mathbb{E}[\|\mathcal{H}(x_k) - x_k + w_k\|_s^2 \mid \mathcal{F}_k]}_{E_2}. \tag{7}
$$

We first control the term $E_1$ in the following. Using the fact that $\mathcal{H}(x^*) = x^*$, we have

$$
E_1 = \underbrace{\langle \nabla M_f^{\mu,g}(x_k - x^*), \mathcal{H}(x_k) - \mathcal{H}(x^*) \rangle}_{E_{1,1}} - \underbrace{\langle \nabla M_f^{\mu,g}(x_k - x^*), x_k - x^* \rangle}_{E_{1,2}}. \tag{8}
$$

For the gradient of $M_f^{\mu,g}(x)$, since $M_f^{\mu,g}(x) = \frac{1}{2}\|x\|_M^2$, we have by the chain rule of subdifferential calculus (Theorem 3.47 of [1]) that $\nabla M_f^{\mu,g}(x) = \|x\|_M v_x$, where $v_x \in \partial\|x\|_M$ is a subgradient of the function $\|x\|_M$ at $x$. In fact, from the equation $\nabla M_f^{\mu,g}(x) = \|x\|_M v_x$, we see that $v_x$ is unique (i.e., $v_x = \nabla\|x\|_M$) for all $x \neq 0$.

Now consider the term $E_{1,1}$. Using Hölder's inequality, we have

$$
\begin{aligned}
E_{1,1} &= \|x_k - x^*\|_M \langle v_{x_k - x^*}, \mathcal{H}(x_k) - \mathcal{H}(x^*) \rangle \\
&\leq \|x_k - x^*\|_M \|v_{x_k - x^*}\|_M^* \|\mathcal{H}(x_k) - \mathcal{H}(x^*)\|_M, \tag{9}
\end{aligned}
$$

where $\|\cdot\|_M^*$ is the dual norm of $\|\cdot\|_M$. To further control $E_{1,1}$, the following result is needed.

**Lemma A.1** (Lemma 2.6 of [36]). *Let $h : \mathcal{D} \to \mathbb{R}$ be a convex function. Then $h$ is $L$ – Lipschitz over $\mathcal{D}$ with respect to norm $\| \cdot \|$ if and only if for all $w \in \mathcal{D}$ and $z \in \partial h(w)$ we have that $\|z\|_* \leq L$, where $\| \cdot \|_*$ is the dual norm of $\| \cdot \|$.*

Since $\|x\|_M$ as a function of $x$ is $1$ – Lipschitz w.r.t. $\|\cdot\|_M$, we have by Lemma A.1 that $\|v_{x_k - x^*}\|_M^* \leq 1$. For the term $\|\mathcal{H}(x_k) - \mathcal{H}(x^*)\|_M$ in Eq. (9), using Lemma 2.1 (b) and the contraction of $\mathcal{H}$ with respect to $\| \cdot \|_c$, we have

$$
\begin{aligned}
\frac{1}{2}\|\mathcal{H}(x_k) - \mathcal{H}(x^*)\|_M^2 &= M_f^{\mu,g}(\mathcal{H}(x_k) - \mathcal{H}(x^*)) \\[2mm]
&\leq \frac{1}{1 + \mu/u_{cs}^2} f(\mathcal{H}(x_k) - \mathcal{H}(x^*)) && \text{(Lemma 2.1 (b))} \\[2mm]
&\leq \frac{\gamma^2}{1 + \mu/u_{cs}^2} f(x_k - x^*) && \text{(Assumption 2.1)} \\[2mm]
&\leq \gamma^2 \frac{1 + \mu/\ell_{cs}^2}{1 + \mu/u_{cs}^2} M_f^{\mu,g}(x_k - x^*) && \text{(Lemma 2.1 (b))} \\[2mm]
&= \frac{\gamma^2}{2}\frac{1 + \mu/\ell_{cs}^2}{1 + \mu/u_{cs}^2}\|x_k - x^*\|_M^2,
\end{aligned}
$$

which implies

$$
\|\mathcal{H}(x_k) - \mathcal{H}(x^*)\|_M \leq \gamma \left( \frac{1 + \mu/\ell_{cs}^2}{1 + \mu/u_{cs}^2} \right)^{1/2} \|x_k - x^*\|_M.
$$

Substituting the upper bounds we obtained for $\|v_{x_k - x^*}\|_M^*$ and $\|\mathcal{H}(x_k) - \mathcal{H}(x^*)\|_M$ into Eq. (9), we have

$$
\begin{aligned}
E_{1,1} &\leq \|x_k - x^*\|_M \|v_{x_k - x^*}\|_M^* \|\mathcal{H}(x_k) - \mathcal{H}(x^*)\|_M \\[2mm]
&\leq \gamma \left( \frac{1 + \mu/\ell_{cs}^2}{1 + \mu/u_{cs}^2} \right)^{1/2} \|x_k - x^*\|_M^2 \\[2mm]
&= 2\gamma \left( \frac{1 + \mu/\ell_{cs}^2}{1 + \mu/u_{cs}^2} \right)^{1/2} M_f^{\mu,g}(x_k - x^*).
\end{aligned}
$$

Now consider the term $E_{1,2}$ in Eq. (8). Since the norm $\| \cdot \|_M$ is a convex function of $x$, we have by definition of convexity that $\|0\|_M - \|x_k - x^*\|_M \geq \langle v_{x_k - x^*}, -(x_k - x^*) \rangle$. Therefore, we have

$$
E_{1,2} = \|x_k - x^*\|_M \langle v_{x_k - x^*}, x_k - x^* \rangle \geq \|x_k - x^*\|_M^2 = 2M_f^{\mu,g}(x_k - x^*).
$$

Combining the bounds on $E_{1,1}$ and $E_{1,2}$, we obtain

$$
E_1 = E_{1,1} - E_{1,2} \leq -2 \left[ 1 - \gamma \left( \frac{1 + \mu/\ell_{cs}^2}{1 + \mu/u_{cs}^2} \right)^{1/2} \right] M_f^{\mu,g}(x_k - x^*).
$$

We next analyze the term $E_2$ in Eq. (7) in the following:

$$
\begin{aligned}
E_2 &= \mathbb{E}\left[ \|\mathcal{H}(x_k) - x_k + w_k\|_s^2 \mid \mathcal{F}_k \right] \\[2mm]
&= \mathbb{E}\left[ \|\mathcal{H}(x_k) - \mathcal{H}(x^*) + x^* - x_k + w_k\|_s^2 \mid \mathcal{F}_k \right] && (\mathcal{H}(x^*) = x^*) \\[2mm]
&\leq \mathbb{E}\left[ \left( \|\mathcal{H}(x_k) - \mathcal{H}(x^*)\|_s + \|x_k - x^*\|_s + \|w_k\|_s \right)^2 \mid \mathcal{F}_k \right] \\[2mm]
&\leq \mathbb{E}\left[ \left( u_{cs}\|\mathcal{H}(x_k) - \mathcal{H}(x^*)\|_c + u_{cs}\|x_k - x^*\|_c + u_{ns}\|w_k\|_n \right)^2 \mid \mathcal{F}_k \right] \\[2mm]
&\leq \mathbb{E}\left[ \left( 2u_{cs}\|x_k - x^*\|_c + u_{ns}\|w_k\|_n \right)^2 \mid \mathcal{F}_k \right] && \text{(Assumption 2.1)} \\[2mm]
&\leq 8u_{cs}^2\|x_k - x^*\|_c^2 + 2u_{ns}^2\mathbb{E}\left[ \|w_k\|_n^2 \mid \mathcal{F}_k \right] && ((a+b)^2 \leq 2(a^2 + b^2)) \\[2mm]
&\leq 8u_{cs}^2\|x_k - x^*\|_c^2 + 2Au_{ns}^2(1 + \|x_k\|_n^2) && \text{(Assumption 2.2 (b))} \\[2mm]
&\leq 8u_{cs}^2\|x_k - x^*\|_c^2 + \frac{2Au_{ns}^2 u_{cs}^2}{\ell_{ns}^2}(1 + \|x_k\|_c^2) && (u_{cs} \geq 1 \text{ and } \ell_{ns} \leq 1)
\end{aligned}
$$

$$\leq 8u_{cs}^2\|x_k - x^*\|_c^2 + \frac{2Au_{ns}^2u_{cs}^2}{\ell_{ns}^2}(1 + 2\|x_k - x^*\|_c^2 + 2\|x^*\|_c^2)$$

$$\leq 4u_{cs}^2\left(2 + \frac{Au_{ns}^2}{\ell_{ns}^2}\right)\|x_k - x^*\|_c^2 + \frac{2Au_{ns}^2u_{cs}^2}{\ell_{ns}^2}(1 + 2\|x^*\|_c^2)$$

$$\leq \frac{8u_{cs}^2u_{ns}^2(A+2)}{\ell_{ns}^2}f(x_k - x^*) + \frac{2Au_{ns}^2u_{cs}^2}{\ell_{ns}^2}(1 + 2\|x^*\|_c^2)$$

$$\leq \frac{8u_{cs}^2u_{ns}^2(A+2)(\ell_{cs}^2 + \mu)}{\ell_{cs}^2\ell_{ns}^2}M_f^{\mu,g}(x_k - x^*) + \frac{2Au_{ns}^2u_{cs}^2}{\ell_{ns}^2}(1 + 2\|x^*\|_c^2). \quad \text{(Lemma 2.1 (b))}$$

Substituting the upper bounds we obtained for the terms $E_1$ and $E_2$ into Eq. (7), we finally have for all $k \geq 0$:

$$\mathbb{E}[M_f^{\mu,g}(x_{k+1} - x^*) \mid \mathcal{F}_k]$$

$$\leq \left\{1 - 2\left[1 - \gamma\left(\frac{1 + \mu/\ell_{cs}^2}{1 + \mu/u_{cs}^2}\right)^{1/2}\right]\epsilon_k + \frac{4u_{cs}^2u_{ns}^2(A+2)L(\ell_{cs}^2 + \mu)}{\mu\ell_{cs}^2\ell_{ns}^2}\epsilon_k^2\right\}M_f^{\mu,g}(x_k - x^*)$$

$$+ \frac{ALu_{ns}^2u_{cs}^2\epsilon_k^2}{\mu\ell_{ns}^2}(1 + 2\|x^*\|_c^2)$$

$$= (1 - 2\alpha_2\epsilon_k + \alpha_3\epsilon_k^2)M_f^{\mu,g}(x_k - x^*) + \frac{\alpha_4(1 + 2\|x^*\|_c^2)}{2(1 + \mu/\ell_{cs}^2)}\epsilon_k^2.$$

## A.3 Proof of Theorem 2.1

We begin with Eq. (3) of Proposition 2.1. When $\epsilon_0 \leq \alpha_2/\alpha_3$, we have by monotonicity of $\{\epsilon_k\}$ that:

$$\mathbb{E}[M_f^{\mu,g}(x_{k+1} - x^*) \mid \mathcal{F}_k] \leq (1 - \alpha_2\epsilon_k)M_f^{\mu,g}(x_k - x^*) + \frac{\alpha_4(1 + 2\|x^*\|_c^2)}{2(1 + \mu/\ell_{cs}^2)}\epsilon_k^2$$

for all $k \geq 0$. Taking the total expectation on both side of the previous inequality and then recursively using it, we obtain

$$\mathbb{E}[M_f^{\mu,g}(x_k - x^*)] \leq \prod_{j=0}^{k-1}(1 - \alpha_2\epsilon_j)M_f^{\mu,g}(x_0 - x^*) + \frac{\alpha_4(1 + 2\|x^*\|_c^2)}{2(1 + \mu/\ell_{cs}^2)}\sum_{i=0}^{k-1}\epsilon_i^2\prod_{j=i+1}^{k-1}(1 - \alpha_2\epsilon_j).$$

The above inequality is the finite-sample bounds of $M_f^{\mu,g}(x_k - x^*)$. To write it in terms of the original norm square $\|x_k - x^*\|_c^2$, using Lemma 2.1 (b) one more time and we finally obtain

$$\mathbb{E}\left[\|x_k - x^*\|_c^2\right] \leq \alpha_1\|x_0 - x^*\|_c^2\prod_{j=0}^{k-1}(1 - \alpha_2\epsilon_j) + \alpha_4(1 + 2\|x^*\|_c^2)\sum_{i=0}^{k-1}\epsilon_i^2\prod_{j=i+1}^{k-1}(1 - \alpha_2\epsilon_j)$$

for all $k \geq 0$.

## A.4 Proof of Corollary 2.1

We begin with Eq. (4) of Theorem 2.1. When $\epsilon_k = \epsilon \leq \alpha_2/\alpha_3$ for all $k \geq 0$, we have

$$\mathbb{E}\left[\|x_k - x^*\|_c^2\right] \leq \alpha_1\|x_0 - x^*\|_c^2\prod_{j=0}^{k-1}(1 - \alpha_2\epsilon_j) + \alpha_4(1 + 2\|x^*\|_c^2)\sum_{i=0}^{k-1}\epsilon_i^2\prod_{j=i+1}^{k-1}(1 - \alpha_2\epsilon_j)$$

$$= \alpha_1\|x_0 - x^*\|_c^2(1 - \alpha_2\epsilon)^k + \alpha_4(1 + 2\|x^*\|_c^2)\epsilon^2\sum_{i=0}^{k-1}(1 - \alpha_2\epsilon)^{k-i-1}$$

$$\leq \alpha_1\|x_0 - x^*\|_c^2(1 - \alpha_2\epsilon)^k + (1 + 2\|x^*\|_c^2)\frac{\alpha_4\epsilon}{\alpha_2}.$$

## A.5 Proof of Corollary 2.2

We begin with Eq. (4)

$$\mathbb{E}\left[\|x_k - x^*\|_c^2\right] \le \alpha_1 \|x_0 - x^*\|_c^2 \underbrace{\prod_{j=0}^{k-1}(1 - \alpha_2\epsilon_j)}_{T_1} + \alpha_4(1 + 2\|x^*\|_c^2) \underbrace{\sum_{i=0}^{k-1} \epsilon_i^2 \prod_{j=i+1}^{k-1}(1 - \alpha_2\epsilon_j)}_{T_2}.$$

We next evaluate the terms $T_1$ and $T_2$ for different values of $\xi$ and $\epsilon$.

### A.5.1 The term $T_1$

Using the expression for $\epsilon_k$ and the relation that $e^x \ge 1 + x$ for all $x \in \mathbb{R}$, we have

$$T_1 = \prod_{j=0}^{k-1}(1 - \alpha_2\epsilon_j) = \prod_{j=0}^{k-1}\left(1 - \frac{\alpha_2\epsilon}{(j+K)^\xi}\right) \le \exp\left(-\alpha_2\epsilon \sum_{i=0}^{k-1} \frac{1}{(j+K)^\xi}\right).$$

Since the inequality $\int_a^{b+1} h(x)dx \le \sum_{n=a}^{b} h(n) \le \int_{a-1}^{b} h(x)dx$ holds for any non-increasing function $h(x)$, we have

$$T_1 \le \exp\left(-\alpha_2\epsilon \int_0^k \frac{1}{(x+K)^\xi}dx\right) \le \begin{cases} \left(\dfrac{K}{k+K}\right)^{\alpha_2\epsilon}, & \xi = 1, \\[2em] \exp\left[-\dfrac{\alpha_2\epsilon}{1-\xi}\left((k+K)^{1-\xi} - K^{1-\xi}\right)\right], & \xi \in (0,1). \end{cases}$$

### A.5.2 The term $T_2$

When $\xi = 1$, using the expression of $\epsilon_k$, we have

$$T_2 = \sum_{i=0}^{k-1} \epsilon_i^2 \prod_{j=i+1}^{k-1}(1 - \alpha_2\epsilon_j)$$

$$= \epsilon^2 \sum_{i=0}^{k-1} \frac{1}{(i+K)^2} \prod_{j=i+1}^{k-1}\left(1 - \frac{\alpha_2\epsilon}{j+K}\right)$$

$$= \epsilon^2 \sum_{i=0}^{k-1} \frac{1}{(i+K)^2} \exp\left(-\alpha_2\epsilon \sum_{j=i+1}^{k-1} \frac{1}{j+K}\right)$$

$$\le \epsilon^2 \sum_{i=0}^{k-1} \frac{1}{(i+K)^2} \exp\left(-\alpha_2\epsilon \int_{i+1}^k \frac{1}{x+K}dx\right)$$

$$\le \epsilon^2 \sum_{i=0}^{k-1} \frac{1}{(i+K)^2} \left(\frac{i+1+K}{k+K}\right)^{\alpha_2\epsilon}$$

$$\le \frac{4\epsilon^2}{(k+K)^{\alpha_2\epsilon}} \sum_{i=0}^{k-1} \frac{1}{(i+1+K)^{2-\alpha_2\epsilon}},$$

where the last line follows from

$$\left(\frac{i+1+K}{i+K}\right)^2 \le \left(\frac{K+1}{K}\right)^2 \le 4.$$

We next consider the quantity

$$\sum_{i=0}^{k-1} \frac{1}{(i+1+K)^{2-\alpha_2\epsilon}},$$

whose upper bounds depend on the relation between $\alpha_2\epsilon$ and 2. Using the same technique as before, i.e., bounding the summation by its corresponding integral, we have the following results.

(1) When $\epsilon \in (0, 1/\alpha_2)$, we have $\sum_{i=0}^{k-1} \frac{1}{(i+1+K)^{2-\alpha_2\epsilon}} \leq \frac{1}{1-\alpha_2\epsilon}$.

(2) When $\epsilon = 1/\alpha_2$, we have $\sum_{i=0}^{k-1} \frac{1}{i+1+K} \leq \log(k+K)$.

(3) When $\epsilon \in (1/\alpha_2, 2/\alpha_2)$, we have $\sum_{i=0}^{k-1} \frac{1}{(i+1+K)^{2-\alpha_2\epsilon}} \leq \frac{1}{\alpha_2\epsilon-1}(k+K)^{\alpha_2\epsilon-1}$.

(4) When $\epsilon = 2/\alpha_2$, we have $\sum_{i=0}^{k-1} \frac{1}{(i+1+K)^0} = k$.

(5) when $\epsilon > 2/\alpha_2$, we have $\sum_{i=0}^{k-1} \frac{1}{(i+1+K)^{2-\alpha_2\epsilon}} \leq \frac{e}{\alpha_2\epsilon-1}(k+K)^{\alpha_2\epsilon-1}$.

Combine the above five cases together and we have when $\xi = 1$:

$$T_2 \leq \frac{4\epsilon^2}{(k+K)^{\alpha_2\epsilon}}\sum_{i=0}^{k-1}\frac{1}{(i+1+K)^{2-\alpha_2\epsilon}} \leq \begin{cases} \dfrac{4\epsilon^2}{1-\alpha_2\epsilon}\dfrac{1}{(k+K)^{\alpha_2\epsilon}}, & \epsilon \in (0, 1/\alpha_2), \\[2mm] 4\epsilon^2\dfrac{\log(k+K)}{k+K}, & \epsilon = 1/\alpha_2, \\[2mm] \dfrac{4e\epsilon^2}{\alpha_2\epsilon-1}\dfrac{1}{k+K}, & \epsilon \in (1/\alpha_2, \infty). \end{cases}$$

When $\xi \in (0, 1)$, the approach we used earlier does not work because the integral we used to bound the sum does not admit a clean analytical expression. Here we present one way to evaluate $T_2$ based on induction. Consider the sequence $\{u_k\}_{k\geq 0}$ defined by

$$u_0 = 0, \quad u_{k+1} = (1-\alpha_2\epsilon_k)u_k + \epsilon_k^2, \quad \forall\, k \geq 0.$$

It can be easily verified that $u_k = \sum_{i=0}^{k-1}\epsilon_i^2\prod_{j=i+1}^{k-1}(1-\alpha_2\epsilon_j) = T_2$. We next use induction on $u_k$ to show that when $k \geq \max(0, [2\xi/(\alpha_2\epsilon)]^{1/(1-\xi)} - K)$, we have $u_k \leq \frac{2\epsilon}{\alpha_2}\frac{1}{(k+K)^\xi}$. Since $u_0 = 0 \leq \frac{2\epsilon}{\alpha_2}\frac{1}{K^\xi}$, we have the base case. Now suppose $u_k \leq \frac{2\epsilon}{\alpha_2}\frac{1}{(k+K)^\xi}$ for some $k > 0$. Consider $u_{k+1}$. We have

$$\frac{2\epsilon}{\alpha_2}\frac{1}{(k+1+K)^\xi} - u_{k+1}$$

$$= \frac{2\epsilon}{\alpha_2}\frac{1}{(k+1+K)^\xi} - (1-\alpha_2\epsilon_k)u_k - \epsilon_k^2$$

$$\geq \frac{2\epsilon}{\alpha_2}\frac{1}{(k+1+K)^\xi} - \left(1 - \frac{\alpha_2\epsilon}{(k+K)^\xi}\right)\frac{2\epsilon}{\alpha_2}\frac{1}{(k+K)^\xi} - \frac{\epsilon^2}{(k+K)^{2\xi}}$$

$$= \frac{2\epsilon}{\alpha_2}\left[\frac{1}{(k+1+K)^\xi} - \frac{1}{(k+K)^\xi} + \frac{\alpha_2\epsilon}{2}\frac{1}{(k+K)^{2\xi}}\right]$$

$$= \frac{2\epsilon}{\alpha_2}\frac{1}{(k+K)^{2\xi}}\left[\frac{\alpha_2\epsilon}{2} - (k+K)^\xi\left(1 - \left(\frac{k+K}{k+1+K}\right)^\xi\right)\right].$$

Note that

$$\left(\frac{k+K}{k+1+K}\right)^\xi = \left[\left(1+\frac{1}{k+K}\right)^{k+K}\right]^{-\frac{\xi}{k+K}} \geq \exp\left(-\frac{\xi}{k+K}\right) \geq 1 - \frac{\xi}{k+K},$$

where we used $(1+\frac{1}{x})^x < e$ for all $x > 0$ and $e^x \geq 1+x$ for all $x \in \mathbb{R}$. Therefore, we obtain

$$\frac{2\epsilon}{\alpha_2}\frac{1}{(k+1+K)^\xi} - u_{k+1} = \frac{2\epsilon}{\alpha_2}\frac{1}{(k+K)^{2\xi}}\left[\frac{\alpha_2\epsilon}{2} - (k+K)^\xi\left(1 - \left(\frac{k+K}{k+1+K}\right)^\xi\right)\right]$$

$$\geq \frac{2\epsilon}{\alpha_2}\frac{1}{(k+K)^{2\xi}}\left[\frac{\alpha_2\epsilon}{2} - \frac{\xi}{(k+K)^{1-\xi}}\right]$$

$$\geq 0,$$

where the last line follows from $K \geq [2\xi/(\alpha_2\epsilon)]^{1/(1-\xi)}$. The induction is now complete, and we have $T_2 \leq \frac{2\epsilon}{\alpha_2}\frac{1}{(k+K)^\xi}$ for all $k \geq 0$.

Finally, combine the results in Subsections A.5.1 and A.5.2, we have the finite-sample error bounds given in Corollary 2.2.

## A.6 Proof of Corollary 2.3

When $\|\cdot\|_c = \|\cdot\|_n = \|\cdot\|_\infty$, we choose the smoothing function as $g(x) = \frac{1}{2}\|x\|_p^2$ with $p \geq 2$. It is known that $g(x)$ is $(p-1)$ − smooth w.r.t. $\|\cdot\|_p$ (Example 5.11 [1]). Moreover, we have in this case $\ell_{cs} = \ell_{ns} = 1$ and $u_{cs} = u_{ns} = d^{1/p}$. It follows that

$$
\alpha_1 = \frac{1+\mu}{1+\mu/d^{2/p}}, \qquad\qquad \alpha_2 = 1 - \gamma\left(\frac{1+\mu}{1+\mu/d^{2/p}}\right)^{1/2},
$$

$$
\alpha_3 = 4d^{4/p}(p-1)(1+1/\mu)(A+2), \qquad \alpha_4 = 2d^{4/p}(p-1)(1+1/\mu)A. \tag{10}
$$

Note that $\alpha_3$ and $\alpha_4$ are both proportional to $d^{4/p}(p-1)$. Let $h(p) = d^{4/p}(p-1)$. Assume without loss of generality that $d \geq 2$, then we have $\min_{p \geq 2} h(p) \leq h(4\log(d)) \leq 4e\log(d)$. Hence, with $p = 4\log(d)$, the dimension dependence of $\alpha_3$ and $\alpha_4$ is $\log(d)$.

Now for the choice of $\mu$, observe that $\alpha_1$ and $\alpha_2$ are in favor of small $\mu$, while $\alpha_3$ and $\alpha_3$ are in favor of large $\mu$. To balance the effect, we choose $\mu = (1/2 + 1/2\gamma)^2 - 1$. Note that this choice of $\mu$ gives

$$
1 + \frac{1}{\mu} = \frac{(1+\gamma)^2}{(\gamma+1)^2 - 4\gamma^2} = \frac{(1+\gamma)^2}{(1-\gamma)(1+3\gamma)} \leq \frac{1+\gamma}{1-\gamma} \leq \frac{2}{1-\gamma}.
$$

Substituting $p = 4\log(d)$ and $\mu = (1/2 + 1/2\gamma)^2 - 1$ into Eq. (10), we obtain

$$
\alpha_1 = \frac{1+\mu}{1+\mu/d^{2/p}} = \sqrt{e}\frac{1+\mu}{\sqrt{e}+\mu} \leq \sqrt{e} \leq \frac{3}{2},
$$

$$
\alpha_2 = 1 - \gamma(\frac{1+\mu}{1+\mu/d^{2/p}})^{1/2} \geq 1 - \gamma(1+\mu)^{1/2} = \frac{1-\gamma}{2},
$$

$$
\alpha_3 = 4(1+1/\mu)(A+2)d^{4/p}(p-1) \leq \frac{32e(A+2)\log(d)}{1-\gamma},
$$

$$
\alpha_4 = 2(1+1/\mu)Ad^{4/p}(p-1) \leq \frac{16eA\log(d)}{1-\gamma}.
$$

# B  Proofs of All Technical Results in Section 3

## B.1  Properties of the V-trace algorithm

The ideas for the proofs in (a) - (d) (and Lemma B.1) are essentially the same as in [18], we include them here for completeness.

(a) Algorithm (5) can be rewritten in the following way:

$$
V_{k+1}(s) = V_k(s) + \epsilon_k \sum_{t=0}^{T} \beta^t c_{0,t-1}\rho_t \left(r_t + \beta V_k(S_{t+1}) - V_k(S_t)\right)
$$

$$
= V_k(s) + \epsilon_k \left\{\sum_{t=0}^{T} \beta^t c_{0,t-1}\rho_t \left(r_t + \beta V_k(S_{t+1}) - V_k(S_t)\right) + V_k(s) - V_k(s)\right\}
$$

$$
= V_k(s) + \epsilon_k \left([\mathcal{H}(V_k)](s) - V_k(s) + w_k(s)\right),
$$

where

$$
[\mathcal{H}(V)](s) = \mathbb{E}_{\pi'}\left[\sum_{t=0}^{T} \beta^t c_{0,t-1}\rho_t \left(r_t + \beta V(S_{t+1}) - V(S_t)\right)\ \middle|\ S_0 = s\right] + V(s),
$$

and

$$
w_k(s) = \sum_{t=0}^{T} \beta^t c_{0,t-1}\rho_t(r_t + \beta V_k(S_{t+1}) - V_k(S_t)) + V_k(s) - [\mathcal{H}(V_k)](s).
$$

Hence we have $V_{k+1} = V_k + \epsilon_k \left(\mathcal{H}(V_k) - V_k + w_k\right)$.

(b) We begin by rewriting the operator $\mathcal{H}$ in the following way:

$$[\mathcal{H}(V)](s) = \mathbb{E}_{\pi'}\left[\sum_{t=0}^{T+1}\beta^{t-1}c_{0,t-2}(\rho_{t-1}r_{t-1} + \beta(\rho_{t-1} - c_{t-1}\rho_t)V(S_t))\right.$$

$$\left. + \beta^{T+1}c_{0,T}\rho_{T+1}V(S_{T+1}) \mid S_0 = s\right].$$

For any $V_1, V_2 : \mathbb{R}^n \mapsto \mathbb{R}$ and $s \in \mathcal{S}$, we have

$$[\mathcal{H}V_1](s) - [\mathcal{H}V_2](s)$$

$$= \mathbb{E}_{\pi'}\left[\sum_{t=0}^{T+1}\beta^t c_{0,t-2}(\rho_{t-1} - c_{t-1}\rho_t)(V_1(S_t) - V_2(S_t)) \mid S_0 = s\right]$$

$$+ \mathbb{E}_{\pi'}\left[\beta^{T+1}c_{0,T}\rho_{T+1}(V_1(S_{T+1}) - V_2(S_{T+1})) \mid S_0 = s\right]$$

$$\leq \sum_{t=0}^{T+1}\beta^t\mathbb{E}_{\pi'}\left[c_{0,t-2}(\rho_{t-1} - c_{t-1}\rho_t)(V_1(S_t) - V_2(S_t)) \mid S_0 = s\right] \tag{11}$$

$$+ \beta^{T+1}\mathbb{E}_{\pi'}\left[c_{0,T}\rho_{T+1} \mid S_0 = s\right]\|V_1 - V_2\|_\infty.$$

Since $\bar{\rho} \geq \bar{c}$, we have $\rho_t \geq c_t$ for all $t$. Therefore, using the Markov property and we have

$$\mathbb{E}_{\pi'}\left[\rho_{t-1} - c_{t-1}\rho_t \mid \mathcal{F}_t\right] \geq c_{t-1}\left(1 - \mathbb{E}_{\pi'}\left[\rho_t \mid \mathcal{F}_t\right]\right)$$

$$\geq c_{t-1}\left(1 - \sum_{b \in \mathcal{A}}\pi'(b|S_t)\frac{\pi(b|S_t)}{\pi'(b|S_t)}\right)$$

$$= 0, \tag{12}$$

where $\mathcal{F}_t$ denotes the $\sigma$-algebra generated by $\{S_0, A_0, ..., S_{t-1}, A_{t-1}, S_t\}$. It follows that

$$\sum_{t=0}^{T+1}\beta^t\mathbb{E}_{\pi'}\left[c_{0,t-2}(\rho_{t-1} - c_{t-1}\rho_t)(V_1(S_t) - V_2(S_t)) \mid S_0 = s\right]$$

$$= \sum_{t=0}^{T+1}\beta^t\mathbb{E}_{\pi'}\left[\mathbb{E}_{\pi'}\left[c_{0,t-2}(\rho_{t-1} - c_{t-1}\rho_t)(V_1(S_t) - V_2(S_t)) \mid \mathcal{F}_t\right] \mid S_0 = s\right]$$

$$= \sum_{t=0}^{T+1}\mathbb{E}_{\pi'}\left[\mathbb{E}_{\pi'}\left[c_{0,t-2}(\rho_{t-1} - c_{t-1}\rho_t) \mid \mathcal{F}_t\right](V_1(S_t) - V_2(S_t)) \mid S_0 = s\right]$$

$$\leq \sum_{t=0}^{T+1}\beta^t\mathbb{E}_{\pi'}\left[\mathbb{E}_{\pi'}\left[c_{0,t-2}(\rho_{t-1} - c_{t-1}\rho_t) \mid \mathcal{F}_t\right] \mid S_0 = s\right]\|V_1 - V_2\|_\infty$$

$$= \sum_{t=0}^{T+1}\beta^t\mathbb{E}_{\pi'}\left[c_{0,t-2}(\rho_{t-1} - c_{t-1}\rho_t) \mid S_0 = s\right]\|V_1 - V_2\|_\infty.$$

Using the previous result in Eq. (11) and we have

$$[\mathcal{H}V_1](s) - [\mathcal{H}V_2](s)$$

$$\leq \sum_{t=0}^{T+1}\beta^t\mathbb{E}_{\pi'}\left[c_{0,t-2}(\rho_{t-1} - c_{t-1}\rho_t)(V_1(S_t) - V_2(S_t)) \mid S_0 = s\right]$$

$$+ \beta^{T+1}\mathbb{E}_{\pi'}\left[c_{0,T}\rho_{T+1} \mid S_0 = s\right]\|V_1 - V_2\|_\infty$$

$$\leq \left\{\sum_{t=0}^{T+1}\beta^t\mathbb{E}_{\pi'}\left[c_{0,t-2}(\rho_{t-1} - c_{t-1}\rho_t) \mid S_0 = s\right]\right.$$

$$\left. + \beta^{T+1}\mathbb{E}_{\pi'}\left[c_{0,T}\rho_{T+1} \mid S_0 = s\right]\right\}\|V_1 - V_2\|_\infty.$$

Switching the role between $V_1$ and $V_2$, then we have by symmetry that

$$|[\mathcal{H}V_1](s) - [\mathcal{H}V_2](s)|$$

$$\leq \underbrace{\left\{ \sum_{t=0}^{T+1} \beta^t \mathbb{E}_{\pi'} \left[ c_{0,t-2}(\rho_{t-1} - c_{t-1}\rho_t) \mid S_0 = s \right] + \beta^{T+1}\mathbb{E}_{\pi'} \left[ c_{0,T}\rho_{T+1} \mid S_0 = s \right] \right\}}_{\gamma(s)}$$

$$\times \|V_1 - V_2\|_\infty.$$

To show that $\mathcal{H}$ is a contraction w.r.t. $\|\cdot\|_\infty$, it remains to show that $\gamma(s) < 1$ for all $s \in \mathcal{S}$. Note that

$$\gamma(s) = \sum_{t=0}^{T+1} \beta^t \mathbb{E}_{\pi'} \left[ c_{0,t-2} \left( \rho_{t-1} - c_{t-1}\rho_t \right) \mid S_0 = s \right] + \beta^{T+1}\mathbb{E}_{\pi'} \left[ c_{0,T}\rho_{T+1} \mid S_0 = s \right]$$

$$= \sum_{t=0}^{T+1} \beta^t \mathbb{E}_{\pi'} \left[ c_{0,t-2}\rho_{t-1} \mid S_0 = s \right] - \sum_{t=0}^{T+1} \beta^t \mathbb{E}_{\pi'} \left[ c_{0,t-1}\rho_t \mid S_0 = s \right]$$

$$+ \mathbb{E}_{\pi'} \left[ \beta^{T+1}c_{0,T}\rho_{T+1} \mid S_0 = s \right]$$

$$= 1 + \sum_{t=0}^{T} \beta^{t+1}\mathbb{E}_{\pi'} \left[ c_{0,t-1}\rho_t \mid S_0 = s \right] - \sum_{t=0}^{T} \beta^t \mathbb{E}_{\pi'} \left[ c_{0,t-1}\rho_t \mid S_0 = s \right]$$

$$= 1 - (1 - \beta) \sum_{t=0}^{T} \beta^t \mathbb{E}_{\pi'} \left[ c_{0,t-1}\rho_t \mid S_0 = s \right]$$

$$\leq 1 - (1 - \beta) \min_{s \in \mathcal{S}} \sum_{t=0}^{T} \beta^t \mathbb{E}_{\pi'} \left[ c_{0,t-1}\rho_t \mid S_0 = s \right]$$

$$= \gamma.$$

Let $\zeta := \min_{s \in \mathcal{S}} \sum_{t=0}^{T} \beta^t \mathbb{E}_{\pi'} \left[ c_{0,t-1}\rho_t \mid S_0 = s \right]$, we next show $\zeta > 0$. Using the coverage assumption (i.e., $\{a \mid \pi(a|s) > 0\} \subseteq \{a \mid \pi'(a|s) > 0\}$ for all $s \in \mathcal{S}$), we have

$$\zeta \geq \min_{s \in \mathcal{S}} \mathbb{E}_{\pi'} \left[ \rho_0 \mid S_0 = s \right]$$

$$= \min_{s \in \mathcal{S}} \sum_{a:\pi'(a|s)>0} \min \left( \bar{\rho}\pi'(a|s), \pi(a|s) \right)$$

$$\geq \min_{s \in \mathcal{S}} \sum_{a:\pi(a|s)>0} \min \left( \bar{\rho}\pi'(a|s), \pi(a|s) \right)$$

$$> 0.$$

It follows that $\gamma = 1 - (1 - \beta)\zeta < 1$.

(c) It is enough to show that $V_{\pi_{\bar{\rho}}}$ is a fixed-point of $\mathcal{H}$, the uniqueness part follows from the Banach fixed-point theorem [15]. For any $t \in [0, T]$, we have

$$\mathbb{E}_{\pi'} \left[ \rho_t \left( r_t + \beta V_{\pi_{\bar{\rho}}}(S_{t+1}) - V_{\pi_{\bar{\rho}}}(S_t) \right) \mid S_t \right]$$

$$= \sum_{a \in \mathcal{A}} \pi'(a|S_t) \min \left( \bar{\rho}, \frac{\pi(a|S_t)}{\pi'(a|S_t)} \right) \left( \mathcal{R}(S_t, a) + \beta \sum_{i' \in \mathcal{S}} P_a(S_t, s')V_{\pi_{\bar{\rho}}}(s') - V_{\pi_{\bar{\rho}}}(S_t) \right)$$

$$= \sum_{a \in \mathcal{A}} \pi_{\bar{\rho}}(a|S_t) \left[ \mathcal{R}(S_t, a) + \beta \sum_{i' \in \mathcal{S}} P_a(S_t, s')V_{\pi_{\bar{\rho}}}(s') - V_{\pi_{\bar{\rho}}}(S_t) \right] \sum_{b \in \mathcal{A}} \min(\bar{\rho}\pi'(b|S_t), \pi(b|S_t))$$

$$= 0,$$

where the last line follows from the Bellman's equation for $V_{\pi_{\bar{\rho}}}$. Therefore, using the tower property of the conditional expectation and the Markov property, we have $\mathcal{H}(V_{\pi_{\bar{\rho}}}) = V_{\pi_{\bar{\rho}}}$, hence $V_{\pi_{\bar{\rho}}}$ is a fixed-point of $\mathcal{H}$.

We next analyze the difference between $V_{\pi_{\bar{\rho}}}$ and $V_\pi$ in terms of $\pi_{\bar{\rho}}$ and $\pi$. We first show in the following lemma that the value function as a function of its corresponding policy is Lipschitz continuous.

**Lemma B.1.** *For any two policies $\pi_1$ and $\pi_2$, their corresponding value functions $V_{\pi_1}$ and $V_{\pi_2}$ satisfy $\|V_{\pi_1} - V_{\pi_2}\|_\infty \le \frac{2}{(1-\gamma)^2}\|\pi_1 - \pi_2\|_\infty$.*

*Proof of Lemma B.1.* Note that for any policy $\pi$, its corresponding value function $V_\pi$ satisfies the following Bellman's equation:

$$V_\pi = R_\pi + \gamma P_\pi V_\pi, \tag{13}$$

where $R_\pi(s) = \sum_{a\in\mathcal{A}} \pi(a|s)\mathcal{R}(s,a)$ for all $s\in\mathcal{S}$, and $P_\pi(s,s') = \sum_{a\in\mathcal{A}} \pi(a|s)P_a(s,s')$ for any $s,s'\in\mathcal{S}$. Using Eq. (13) for $V_{\pi_1}$ and $V_{\pi_2}$, and we have

$$
\begin{aligned}
R_{\pi_1} - R_{\pi_2} &= (I - \gamma P_{\pi_1})V_{\pi_1} - (I - \gamma P_{\pi_2})V_{\pi_2}\\
&= (I - \gamma P_{\pi_1})V_{\pi_1} - (I - \gamma P_{\pi_1})V_{\pi_2} + (I - \gamma P_{\pi_1})V_{\pi_2} - (I - \gamma P_{\pi_2})V_{\pi_2}\\
&= (I - \gamma P_{\pi_1})(V_{\pi_1} - V_{\pi_2}) - \gamma(P_{\pi_1} - P_{\pi_2})V_{\pi_2}.
\end{aligned}
$$

Since the matrix $I - \gamma P_\pi$ is invertible for any policy $\pi$ [6], we have

$$
\begin{aligned}
\|V_{\pi_1} - V_{\pi_2}\|_\infty &= \|(I - \gamma P_{\pi_1})^{-1}[(R_{\pi_1} - R_{\pi_2}) + \gamma(P_{\pi_1} - P_{\pi_2})V_{\pi_2}]\|_\infty\\
&\le \|(I - \gamma P_{\pi_1})^{-1}\|_\infty\,[\|R_{\pi_1} - R_{\pi_2}\|_\infty + \gamma\|P_{\pi_1} - P_{\pi_2}\|_\infty\|V_{\pi_2}\|_\infty]. \tag{14}
\end{aligned}
$$

We next control all the terms on the r.h.s. of the preceding inequality. For the term $\|(I - \gamma P_{\pi_1})^{-1}\|_\infty$, we have by definition of the matrix sup-norm that

$$
\begin{aligned}
\|(I - \gamma P_{\pi_1})^{-1}\|_\infty^{-1} &= \inf_{\|x\|_\infty = 1} \|(I - \gamma P_{\pi_1})x\|_\infty\\
&\ge \inf_{\|x\|_\infty = 1} \|x\|_\infty - \gamma\|P_{\pi_1}x\|_\infty\\
&= 1 - \gamma \sup_{\|x\|_\infty = 1} \|P_{\pi_1}x\|_\infty\\
&= 1 - \gamma.
\end{aligned}
$$

It follows that $\|(I - \gamma P_{\pi_1})^{-1}\|_\infty \le \frac{1}{1-\gamma}$. Next, for the term $\|R_{\pi_1} - R_{\pi_2}\|_\infty$, we have

$$
\begin{aligned}
\|R_{\pi_1} - R_{\pi_2}\|_\infty &= \max_{s\in\mathcal{S}} |R_{\pi_1}(s) - R_{\pi_2}(s)|\\
&= \max_{s\in\mathcal{S}} \Big|\sum_{a\in\mathcal{A}} (\pi_1(a|s) - \pi_2(a|s))\mathcal{R}(s,a)\Big|\\
&\le \max_{s\in\mathcal{S}} \sum_{a\in\mathcal{A}} |\pi_1(a|s) - \pi_2(a|s)| \qquad\qquad (\mathcal{R}(s,a)\in[0,1])\\
&= \|\pi_1 - \pi_2\|_\infty.
\end{aligned}
$$

Finally we consider the term $\|P_{\pi_1} - P_{\pi_2}\|_\infty\|V_{\pi_2}\|_\infty$. It is clear that $\|V_{\pi_2}\|_\infty \le \sum_{k=0}^\infty \gamma^k = \frac{1}{1-\gamma}$. For $\|P_{\pi_1} - P_{\pi_2}\|_\infty$, we have

$$
\begin{aligned}
\|P_{\pi_1} - P_{\pi_2}\|_\infty &= \max_{s\in\mathcal{S}} \sum_{s'\in\mathcal{S}} |P_{\pi_1}(s,s') - P_{\pi_2}(s,s')|\\
&= \max_{s\in\mathcal{S}} \sum_{s'\in\mathcal{S}} \Big|\sum_{a\in\mathcal{A}} (\pi_1(a|s) - \pi_2(a|s))P_a(s,s')\Big|\\
&\le \max_{s\in\mathcal{S}} \sum_{s'\in\mathcal{S}} \sum_{a\in\mathcal{A}} |(\pi_1(a|s) - \pi_2(a|s))P_a(s,s')|\\
&= \max_{s\in\mathcal{S}} \sum_{a\in\mathcal{A}} \sum_{s'\in\mathcal{S}} |(\pi_1(a|s) - \pi_2(a|s))P_a(s,s')|\\
&= \max_{s\in\mathcal{S}} \sum_{a\in\mathcal{A}} |\pi_1(a|s) - \pi_2(a|s)|\\
&= \|\pi_1 - \pi_2\|_\infty.
\end{aligned}
$$

Using the upper bounds we obtained for the terms on the r.h.s. of Eq. (14), we have

$$\|V_{\pi_1} - V_{\pi_2}\|_\infty \le \|(I - \gamma P_{\pi_1})^{-1}\|_\infty\,[\|R_{\pi_1} - R_{\pi_2}\|_\infty + \gamma\|P_{\pi_1} - P_{\pi_2}\|_\infty\|V_{\pi_2}\|_\infty]$$

$$\leq \frac{1}{1-\gamma} \left[ \|\pi_1 - \pi_2\|_\infty + \frac{1}{1-\gamma} \|\pi_1 - \pi_2\|_\infty \right]$$

$$\leq \frac{2}{(1-\gamma)^2} \|\pi_1 - \pi_2\|_\infty.$$

$\square$

Using Lemma B.1 on $V_{\pi_{\bar\rho}}$ and $V_\pi$, we have

$$\|V_{\pi_{\bar\rho}} - V_\pi\|_\infty \leq \frac{2}{(1-\gamma)^2} \|\pi_{\bar\rho} - \pi\|_\infty.$$

As a remark, combining the previous inequality with Theorem 3.1 gives

$$\mathbb{E}\left[\|V_k - V_\pi\|_\infty^2\right]$$
$$\leq 2\mathbb{E}\left[\|V_k - V_{\pi_{\bar\rho}}\|_\infty^2\right] + 2\|V_\pi - V_{\pi_{\bar\rho}}\|_\infty^2$$
$$\leq 2048 e^2 (\|V_0 - V_{\pi_{\bar\rho}}\|_\infty^2 + 2\|V_{\pi_{\bar\rho}}\|_\infty^2 + 1)\frac{(A+2)\log|\mathcal{S}|}{(1-\gamma)^3}\frac{1}{k+K} + \frac{8}{(1-\gamma)^4}\|\pi_{\bar\rho} - \pi\|_\infty^2.$$

(d) In the setting of Algorithm (5), $\mathcal{F}_k$ contains all the information in the first $(k-1)$ sets of trajectories. Since the $k$-th set of trajectories are generated independent of the previous ones, conditioning on $\mathcal{F}_k$ simply means that $V_k$ is given. Therefore, by definition of $\{w_k\}$, we have $\mathbb{E}[w_k \mid \mathcal{F}_k] = \mathcal{H}(V_k) - \mathcal{H}(V_k) = 0$. Moreover, we have for all $s \in \mathcal{S}$:

$$|w_k(s)| = \left| \sum_{t=0}^{T} \beta^t c_{0,t-1} \rho_t \left(r_t + \beta V_k(S_{t+1}) - V_k(S_t)\right) + V_k(s) - [\mathcal{H}(V_k)](s) \right|$$

$$\leq 2 \sum_{t=0}^{T} (\beta\bar{c})^t \bar{\rho} \left(1 + (\beta+1)\|V_k\|_\infty\right)$$

$$\leq 4\bar{\rho}(1 + \|V_k\|_\infty) \sum_{t=0}^{T} (\beta\bar{c})^t$$

$$\leq \begin{cases} 4\bar{\rho}(1 + \|V_k\|_\infty)\dfrac{1}{1-\beta\bar{c}}, & \text{when } \beta\bar{c} < 1, \\[2ex] 4\bar{\rho}(1 + \|V_k\|_\infty)(T+1), & \text{when } \beta\bar{c} = 1, \\[2ex] 4\bar{\rho}(1 + \|V_k\|_\infty)\dfrac{(\beta\bar{c})^{T+1}}{\beta\bar{c}-1}, & \text{when } \beta\bar{c} > 1. \end{cases}$$

Therefore, we have $\mathbb{E}_{\pi'}\left[\|w_k\|_\infty^2 \mid \mathcal{F}_k\right] \leq A(1 + \|V_k\|_\infty^2)$, where

$$A = \begin{cases} \dfrac{32\bar{\rho}^2}{(1-\beta\bar{c})^2}, & \text{when } \beta\bar{c} < 1, \\[2ex] 32\bar{\rho}^2(T+1)^2, & \text{when } \beta\bar{c} = 1, \\[2ex] \dfrac{32\bar{\rho}^2(\beta\bar{c})^{2T+2}}{(\beta\bar{c}-1)^2}, & \text{when } \beta\bar{c} > 1. \end{cases}$$

## B.2 Proof of Theorem 3.1

Since we have in this case $\|\cdot\|_c = \|\cdot\|_n = \|\cdot\|_\infty$, Corollary 2.3 is applicable. Let $g(x) = \frac{1}{2}\|x\|_p^2$ with $p = 4\log|\mathcal{S}|$, and let $\mu = (1/2 + 1/(2\gamma))^2 - 1$. Then we have

$$\alpha_1 \leq \frac{3}{2} := \bar{\alpha}_1, \qquad\qquad \alpha_2 \geq \frac{1-\gamma}{2} := \bar{\alpha}_2$$

$$\alpha_3 \leq \frac{32e(A+2)\log|\mathcal{S}|}{1-\gamma} := \bar{\alpha}_3, \qquad\qquad \alpha_4 \leq \frac{16eA\log|\mathcal{S}|}{1-\gamma} := \bar{\alpha}_4.$$

Using Theorem 2.1, with $\epsilon_k = \epsilon/(k+K)$, we have that

$$\mathbb{E}\left[\|V_k - V_{\pi_{\bar\rho}}\|_\infty^2\right]$$

$$\leq \alpha_1 \|V_0 - V_{\pi_{\bar{\rho}}}\|_\infty^2 \prod_{j=0}^{k-1}(1-\alpha_2\epsilon_j) + \alpha_4(1+2\|V_{\pi_{\bar{\rho}}}\|_\infty^2)\sum_{i=0}^{k-1}\epsilon_i^2\prod_{j=i+1}^{k-1}(1-\alpha_2\epsilon_j)$$

$$\leq \bar{\alpha}_1 \|V_0 - V_{\pi_{\bar{\rho}}}\|_\infty^2 \prod_{j=0}^{k-1}(1-\bar{\alpha}_2\epsilon_j) + \bar{\alpha}_4(1+2\|V_{\pi_{\bar{\rho}}}\|_\infty^2)\sum_{i=0}^{k-1}\epsilon_i^2\prod_{j=i+1}^{k-1}(1-\bar{\alpha}_2\epsilon_j).$$

Now, using the same proof that leads us from Theorem 2.1 to Corollary 2.2 with $\alpha_1$ to $\alpha_4$ replaced by $\bar{\alpha}_1$ to $\bar{\alpha}_4$, we have when $\xi = 1$, $\epsilon = 2/\bar{\alpha}_2$, and $K = \bar{\alpha}_3/\bar{\alpha}_2$ (the third case of Corollary 2.2):

$$\mathbb{E}\left[\|V_k - V_{\pi_{\bar{\rho}}}\|_\infty^2\right] \leq \bar{\alpha}_1\|V_0 - V_{\pi_{\bar{\rho}}}\|_\infty^2 \left(\frac{K}{k+K}\right)^{\epsilon\bar{\alpha}_2} + \frac{4e\epsilon^2\bar{\alpha}_4}{\bar{\alpha}_2\epsilon - 1}\frac{(1+2\|V_{\pi_{\bar{\rho}}}\|_\infty^2)}{k+K}$$

$$\leq \left(\frac{2\bar{\alpha}_1\bar{\alpha}_3\|V_0 - V_{\pi_{\bar{\rho}}}\|_\infty^2 + 16e\bar{\alpha}_4(1+2\|V_{\pi_{\bar{\rho}}}\|_\infty^2)}{\bar{\alpha}_2^2}\right)\frac{1}{k+K}.$$

Since

$$\frac{2\bar{\alpha}_1\bar{\alpha}_3\|V_0 - V_{\pi_{\bar{\rho}}}\|_\infty^2 + 16e\bar{\alpha}_4(1+2\|V_{\pi_{\bar{\rho}}}\|_\infty^2)}{\bar{\alpha}_2^2}$$

$$= \frac{4}{(1-\gamma)^2}\left[\frac{96e(A+2)\log|\mathcal{S}|\|V_0 - V_{\pi_{\bar{\rho}}}\|_\infty^2}{(1-\gamma)} + \frac{256e^2A\log|\mathcal{S}|(1+2\|V_{\pi_{\bar{\rho}}}\|_\infty^2)}{(1-\gamma)}\right]$$

$$\leq \frac{1024e^2(A+2)\log|\mathcal{S}|(\|V_0 - V_{\pi_{\bar{\rho}}}\|_\infty^2 + 2\|V_{\pi_{\bar{\rho}}}\|_\infty^2 + 1)}{(1-\gamma)^3},$$

we have for all $k \geq 0$:

$$\mathbb{E}\left[\|V_k - V_{\pi_{\bar{\rho}}}\|_\infty^2\right] \leq 1024e^2(\|V_0 - V_{\pi_{\bar{\rho}}}\|_\infty^2 + 2\|V_{\pi_{\bar{\rho}}}\|_\infty^2 + 1)\frac{(A+2)\log|\mathcal{S}|}{(1-\gamma)^3}\frac{1}{k+K}.$$

## C  Finite-sample analysis of $Q$-learning

We begin by introducing the $Q$-function and the $Q$-learning algorithm [47]. Define the optimal state-action value function $Q^* : \mathcal{S} \times \mathcal{A} \mapsto \mathbb{R}$ by

$$Q^*(s,a) = \mathbb{E}_{\pi^*}\left[\sum_{k=0}^\infty \beta^k r_k \ \middle| \ S_0 = s, A_0 = a\right]$$

for all state-action pairs $(s,a)$, where $A_k$ is sampled from the optimal policy $\pi^*(\cdot|S_k)$ for all $k \geq 1$. It was shown that $Q^*$ uniquely verifies the following Bellman's equation [6, 5]:

$$Q^*(s,a) = \mathcal{R}(s,a) + \beta\mathbb{E}\left[\max_{a'\in\mathcal{A}}Q^*(s',a') \ \middle| \ s,a\right], \quad \forall\,(s,a).$$

Moreover, we have $\pi^*(s) \in \arg\max_{a\in\mathcal{A}}Q^*(s,a)$. Since the optimal policy $\pi^*$ can be directly computed based the optimal $Q$-function, it is enough to estimate $Q^*$, which is done by the $Q$-learning algorithm.

Consider the following $Q$-learning algorithm (in the synchronous setting) of [47]: first initialize $Q_0 \in \mathbb{R}^{|\mathcal{S}||\mathcal{A}|}$, then at each time step, sample from each state-action pair $(s,a)$ its successor state $s'$, and update the estimate $Q_k$ of $Q^*$ according to

$$Q_{k+1}(s,a) = Q_k(s,a) + \epsilon_k\left(r_k + \beta\max_{a'\in\mathcal{A}}Q_k(s',a') - Q_k(s,a)\right), \quad \forall\,(s,a). \qquad (15)$$

Let $\mathcal{H} : \mathbb{R}^{|\mathcal{S}||\mathcal{A}|} \mapsto \mathbb{R}^{|\mathcal{S}||\mathcal{A}|}$ be defined by: $[\mathcal{H}(Q)](s,a) = \mathcal{R}(s,a) + \beta\mathbb{E}[\max_{a'\in\mathcal{A}}Q(s',a') \mid s,a]$ for any function $Q : \mathbb{R}^{|\mathcal{S}||\mathcal{A}|} \mapsto \mathbb{R}$. We can rewrite Algorithm (15) in the vector form as

$$Q_{k+1} = Q_k + \epsilon_k\left(\mathcal{H}(Q_k) - Q_k + w_k\right),$$

where $w_k(s,a) = \mathcal{R}(S_k, A_k) + \beta \max_{a' \in \mathcal{A}} Q_k(s', a') - [\mathcal{H}(Q_k)](s,a)$. We next show the contraction property of $\mathcal{H}$. For any $Q_1, Q_2 : \mathbb{R}^{|\mathcal{S}||\mathcal{A}|} \mapsto \mathbb{R}$, we have for any state-action pairs $(s,a)$:

$$|[\mathcal{H}(Q_1)](s,a) - [\mathcal{H}(Q_2)](s,a)| \leq \beta \mathbb{E}\left[\left|\max_{a' \in \mathcal{A}} Q_1(s',a') - \max_{a' \in \mathcal{A}} Q_2(s',a')\right| \;\middle|\; s,a\right]$$

$$\leq \beta \mathbb{E}\left[\max_{a' \in \mathcal{A}} |Q_1(s',a') - Q_2(s',a')| \;\middle|\; s,a\right]$$

$$\leq \beta \|Q_1 - Q_2\|_\infty.$$

Hence $\mathcal{H}$ is a $\beta$-contraction w.r.t. $\|\cdot\|_\infty$. The fact that $Q^*$ is the unique fixed-point of $\mathcal{H}$ follows from the Bellman's equation for $Q^*$. For the noise $\{w_k\}$, due to the Markov property we have $\mathbb{E}[w_k \mid \mathcal{F}_k] = 0$, where $\mathcal{F}_k$ contains all the information up to the $k$-th iteration. Moreover, since $|w_k(s,a)| \leq 2\beta \|Q_k\|_\infty \leq 2(1 + \|Q_k\|_\infty)$, we have $\mathbb{E}\left[\|w_k\|_\infty^2 \mid \mathcal{F}_k\right] \leq 8(1 + \|Q_k\|_\infty^2)$. To summarize, the $Q$-learning algorithm has the following properties:

(a) Algorithm (15) can be rewritten (in vector form) as $Q_{k+1} = Q_k + \epsilon_k (\mathcal{H}(Q_k) - Q_k + w_k)$.

(b) The mapping $\mathcal{H}$ is a $\beta$-contraction w.r.t. $\|\cdot\|_\infty$ with unique fixed-point $Q^*$.

(c) $\{w_k\}$ satisfies Assumption 2.2 with $\|\cdot\|_n = \|\cdot\|_\infty$ and $A = 8$.

Therefore, Assumptions 2.1 and 2.2 are satisfied and Theorem 2.1 is applicable. To compare our result with existing literature [3, 46], we will apply Corollary 2.1 and Corollary 2.2 case (c) (which gives the optimal asymptotic rate) to obtain the finite-sample error bounds for $Q$-learning.

**Theorem C.1.** *Consider $\{Q_k\}$ of Algorithm (15). Suppose that $\epsilon_k = \epsilon \leq \frac{(1-\beta)^2}{640e \log(|\mathcal{S}||\mathcal{A}|)}$ for all $k \geq 0$. Then we have for all $k \geq 0$:*

$$\mathbb{E}\left[\|Q_k - Q^*\|_\infty^2\right] \leq \frac{3}{2}\|Q_0 - Q^*\|_\infty^2 \left[1 - \frac{(1-\beta)\epsilon}{2}\right]^k + (1 + 2\|Q^*\|_\infty^2)\frac{256e \log(|\mathcal{S}||\mathcal{A}|)\epsilon}{(1-\beta)^2}.$$

*Proof of Theorem C.1.* Using Corollary 2.3, letting $g(x) = \frac{1}{2}\|x\|_p^2$ with $p = 4\log(|\mathcal{S}||\mathcal{A}|)$ and $\mu = (1/2 + 1/(2\beta))^2 - 1$, we have in this problem

$$\alpha_1 \leq \frac{3}{2} := \bar{\alpha}_1, \qquad\qquad \alpha_2 \geq \frac{1-\beta}{2} := \bar{\alpha}_2$$

$$\alpha_3 \leq \frac{320e \log(|\mathcal{S}||\mathcal{A}|)}{1-\beta} := \bar{\alpha}_3, \qquad\qquad \alpha_4 \leq \frac{128e \log(|\mathcal{S}||\mathcal{A}|)}{1-\beta} := \bar{\alpha}_4.$$

Applying Corollary 2.1 with $\epsilon \leq \bar{\alpha}_2/\bar{\alpha}_3 \leq \alpha_2/\alpha_3$, we have

$$\mathbb{E}\left[\|Q_k - Q^*\|_\infty^2\right] \leq \alpha_1 \|Q_0 - Q^*\|_\infty^2 (1 - \alpha_2\epsilon)^k + \alpha_4\epsilon(1 + 2\|Q^*\|_\infty^2)/\alpha_2$$

$$\leq \bar{\alpha}_1 \|Q_0 - Q^*\|_\infty^2 (1 - \bar{\alpha}_2\epsilon)^k + \bar{\alpha}_4\epsilon(1 + 2\|Q^*\|_\infty^2)/\bar{\alpha}_2$$

$$= \frac{3}{2}\|Q_0 - Q^*\|_\infty^2 \left[1 - \frac{1}{2}(1-\beta)\epsilon\right]^k + \frac{256e \log(|\mathcal{S}||\mathcal{A}|)(1 + 2\|Q^*\|_\infty^2)}{(1-\beta)^2}\epsilon.$$

$\square$

Theorem C.1 agrees with [3] (Theorem 2.1) in that the iterates $\{Q_k\}$ converge exponentially fast to a ball centered at $Q^*$ with radius proportional to the stepsize $\epsilon$. However, with our approach, we get the dimensional dependence that scales as $\log(|\mathcal{S}||\mathcal{A}|)$ as compared to $(|\mathcal{S}||\mathcal{A}|)^2$ in [3].

We next consider using diminishing stepsizes.

**Theorem C.2.** *Consider $\{Q_k\}$ of Algorithm (15). Suppose that $\epsilon_k = \epsilon/(k + K)$ with $\epsilon = \frac{4}{1-\beta}$ and $K = \frac{640e \log(|\mathcal{S}||\mathcal{A}|)}{(1-\beta)^3}$. Then we have for all $k \geq 0$:*

$$\mathbb{E}\left[\|Q_k - Q^*\|_\infty^2\right] \leq 8192e^2(1 + 2\|Q^*\|_\infty^2 + \|Q_0 - Q^*\|_\infty^2)\frac{\log(|\mathcal{S}||\mathcal{A}|)}{(1-\beta)^3}\frac{1}{k+K}.$$

*Proof of Theorem C.2.* With the same choices of the function $g(x)$ and $\mu$ given in the proof of Theorem C.1, we have

$$\alpha_1 \leq \frac{3}{2} := \bar{\alpha}_1, \qquad\qquad\qquad \alpha_2 \geq \frac{1-\beta}{2} := \bar{\alpha}_2$$

$$\alpha_3 \leq \frac{320e\log(|\mathcal{S}||\mathcal{A}|)}{1-\beta} := \bar{\alpha}_3, \qquad\qquad \alpha_4 \leq \frac{128e\log(|\mathcal{S}||\mathcal{A}|)}{1-\beta} := \bar{\alpha}_4.$$

Therefore, when $\epsilon_k = \epsilon/(k+K)$ with $\epsilon = \frac{2}{\bar{\alpha}_2} = \frac{4}{1-\beta}$ and $K = \frac{\bar{\alpha}_3}{\bar{\alpha}_2} = \frac{640e\log|\mathcal{S}||\mathcal{A}|}{(1-\beta)^3}$, we have by Corollary 2.2 (c) that

$$\mathbb{E}\left[\|Q_k - Q^*\|_\infty^2\right] \leq \alpha_1 \|Q_0 - Q^*\|_\infty^2 \left(\frac{K}{k+K}\right)^{\alpha_2\epsilon} + \frac{4e\epsilon^2\alpha_4(1+2\|Q^*\|_\infty^2)}{\alpha_2\epsilon - 1}\frac{1}{k+K}$$

$$\leq \bar{\alpha}_1 \|Q_0 - Q^*\|_\infty^2 \left(\frac{K}{k+K}\right)^{\bar{\alpha}_2\epsilon} + \frac{4e\epsilon^2\bar{\alpha}_4(1+2\|Q^*\|_\infty^2)}{\bar{\alpha}_2\epsilon - 1}\frac{1}{k+K}$$

$$\leq 8192e^2(1+2\|Q^*\|_\infty^2 + \|Q_0 - Q^*\|_\infty^2)\frac{\log(|\mathcal{S}||\mathcal{A}|)}{(1-\beta)^3}\frac{1}{k+K}.$$

$\square$

The error bound in Theorem C.2 is similar to Corollary 3 of [46] where the dimension dependence appears as $\log(|\mathcal{S}||\mathcal{A}|)$ in the bound and the rate of convergence is $O(1/k)$. However, to derive such result, besides a similar contraction property of $\mathcal{H}$, [46] also used the monotonicity property of $\mathcal{H}$, and the fact that the iterates of $Q$-learning are uniformly bounded. Therefore, our approach is more general in that we need only the contraction property, and weaker noise assumptions.