[Reviews · NeurIPS 2020]

Review 1

Summary and Contributions: The paper studies stochastic approximation (SA) methods in which the update operator is a pseudo-contraction with respect to an arbitrary norm and the noise is a martingale difference sequence with a state dependent bound on the variance. By using Generalized Moreau Envelopes, the paper deduces finite sample bounds for the convergence of such SA methods, more precisely, for the expected distance of the iterates from the fixed point of the update operator. Both the constant step size (learning rate) and the diminishing step size cases are considered. It is also shown that, for the sup norm case, the dependence on the dimension of the state space is logarithmic. Finally, the results are applied to the V-trace method, an off-policy reinforcement learning (RL) algorithm.

Strengths: Stochastic approximation is a very important topic for machine learning, as it lays the theoretical foundation of several learning methods. The classical theory of SA concentrates on asymptotic results, while there is an increasing interest in providing finite sample guarantees for learning algorithms. The paper contributes to this project by deriving bounds for the case when the update operator consist of a pseudo-contraction mapping. This case is especially important for reinforcement learning (RL), as it is well-known that the optimal value function is a fixed point of the Bellman operator which is a contraction (this fact is exploited by several RL methods, e.g., Q-learning). An advantage of the paper is that the contraction can be with respect to any norm, and if it is w.r.t. the sup norm then the dependence on the dimension is logarithmic. The paper is well-written, the method and the needed assumptions are adequately highlighted and discussed, it contains a literature review, it provides solid theoretical results and demonstrates the importance of the general theorems on a specific RL example: the (off-policy) V-trace algorithm. The results are novel, to the best of my knowledge.

Weaknesses: Weaknesses of the paper are: (1) There are no experimental evaluations. The authors should have included simulation experiments on standard RL problems about the V-trace method: comparing its empirical performance with the obtained bound. Such experiments could have helped judging how conservative the obtained bounds are. (2) The results are only valid for the case of pseudo-contraction updates. This is important for RL, but most SA methods do not have this property. (3) The paper only considers the case of martingale difference noises, which is restrictive. Obtaining bounds for more general processes would increase the significance of the results. (4) The obtained results for RL are only valid for MDPs with finite state and action spaces. (5) The title of the paper is a bit too general, it should mention that only contraction based SA methods are considered. Also, the abstract should highlight that only martingale difference type noises are considered.

Correctness: The obtained results seem correct.

Clarity: Yes, the paper is well-written. The analysed method and the applied assumptions are clearly described. There are explanations and an adequate literature review.

Relation to Prior Work: Yes, the authors know the literature well and discuss the relation of their results to the previous contributions.

Reproducibility: Yes

Additional Feedback: Please, change the title to indicate that only contraction based SA methods are considered.


Review 2

Summary and Contributions: This paper considers the fundamental problem of computing a fixed point of a stochastic contractive operator. Formally, this paper supposes that there is an operator H from R^d to R^d such that ||H(x) – x_*||_c \leq \gamma ||x – x_*||_c for some norm || \cdot ||_c, some gamma \in (0, 1), and fixed _x_* and that given any point x can compute H(x) + w where w is mean zero noise, the magnitude of which depends on x. The paper then considers the natural stochastic approximation (SA) algorithm x_k+1 = x_k + epsilon_k (H(x_k) – x_k + w_k) for some sequences step sizes epsilon_k and stochastic mean-0 w_k, the norm of which depends on x_k. The paper provides general convergence bounds for this algorithm in terms of the norm in which H contracts and the bound on the noise vector w_k in terms of x_k. Formally, the paper supposes that conditioned up to iteration k, E||w_k||^2_n \leq A (1 + ||x_k||_n^2) for some norm and provides bounds on the convergence of the method. Further, the paper then leverages this analysis to analyze certain TD-learning algorithms. The paper achieves its result by considering another norm || \cdot ||_s such that g(x) = (1/2) || x||_s^2 is smooth with respect to || \cdot ||_s. The paper considers a smoothing of || \cdot ||_c by norm || \cdot ||_s by the generalized Moreau envelope and uses this to give a potential function to analyze the smooth approximation algorithm. The paper leverages this to give bounds of convergence of the SA algorithm in terms of the smoothness of || \cdot ||_s, how well these norms approximate each other, the choice of A and epsilon_k, and parameters of the smoothing. Leveraging this analysis, the paper gives the analysis of different step size sequences under different assumptions and analyzes the particular case where || \cdot ||_c is the infinity norm. Further, the paper then applies this analysis to analyze the V-trace algorithm for certain reinforcement learning algorithms. In particular, the paper considers a particular procedure for estimating values of a policy in Markov decision process using the trajectory generated by another policy. the paper shows that the algorithm for estimating values can be written as an instance of the proposed stochastic approximation algorithm and analyzes it. UPDATE AFTER AUTHOR FEEDBACK AND DISCUSSION: Thank you to the authors for their thoughtful response. After reading the response and further discussion my core view of the paper is similar to what was originally expressed in this review. I think this paper provides an interesting general result and technique for analyzing SA, however further work to demonstrate the novelty of the analysis or its utility for well-studied theoretical or practical problems would be beneficial. Even though the parameter choice for V-trace mentioned were considered before, I still think it would be beneficial to possibly simplify the assumptions or show that the SA analysis applies to well-studied RL settings to give further bounds. [Also, in response to the comment about “simple ways to analyze SA with unbounded noise.” I agree the setting I mentioned is different than the one considered in the paper and easier to analyze. However, I mentioned it because I think it provides further context on claims about the first analysis of SA of unbounded affine noise. In short, think it would be good to emphasize that the affine part of the statement is critical.]

Strengths: This paper considers a natural algorithm for computing fixed points and provides a nice analysis of its convergence. The approach taken to analyze the method is very principled and clean and the bounds achieved are very general. The paper argues that this is the first finite sample of the proposed method where the noise can be unbounded and the paper provides a variety of applications of this method, e.g. the case of ell_infinity norm and policy evaluation. Further, in the case of ell_infinity norm the paper analyzes a number of step sizes choices and uses that to further clarify the complexity of the problem. Given the prevalence of stochastic fixed-point problems and reinforcement learning problems, this work is of interest to the broader NeurIPS community and could provide a natural benchmark for further improvements. This paper provides a very general set of analysis tools for these fundamental problems and consequently could find further use. Further, as the optimal complexity of the problems considered seems to be unknown, this paper could encourage further work in this area.

Weaknesses: As explained in the summary, the paper makes a number of assumptions about the SA algorithm in order to achieve their guarantees. While the resulting general analysis the paper achieves seems interesting, given all these assumptions made and the lack of prior work, it is difficult to ascertain the ultimate difficulty in attaining these bounds and the novelty of the resulting analysis. Smoothing the objective and analyzing the resulting error seems quite natural and it would be beneficial if the paper could provide further evidence of the novelty of the analysis and applications or argue why the assumptions made are canonical. Further, it is unclear how optimal the analysis of the algorithm is and it would be beneficial to comment on this as well. Further, the paper argues that it is the first to analyze SA with unbounded errors, however it seems that there are simple, but naturally occurring cases of SA, where the error scales with the norm of the iterate and yet it is simple to analyze SA. For example, suppose that ||w_k||_c \leq c ||x_k||_c for some sufficiently small constant c, dependent on gamma. Here, it seems straightforward to argue that each iteration of SA contracts even the size of the error depends on the iterate (but won’t grow as the algorithm continues). Moreover, this is not very far-off from what happens in certain algorithms for RL-related problems, e.g. computing policies in MDPs by sampling and (by Jensen’s inequality), up to some additive error and change in norm, not completely dissimilar from the setting considered. Consequently, it would be beneficial to comment on this and compare to this approach. Finally, while the application to TD-learning is interesting, again there are many assumptions made for this analysis and therefore gauging the novelty of this result is difficult for the same reasons given above. Further, it seems that the paper is analyzing a particular policy evaluation algorithm, rather than giving a complete analysis of a reinforcement learning algorithm and problem. It would be beneficial to have the writing clarify this a little earlier. Lastly, for both SA and TD-learning, it seems that prior work did provide convergence results. It would be beneficial to provide a little more comparison between the analysis provided here and the analysis in these papers.

Correctness: As far as I can tell the claims are correct, though I have not verified them.

Clarity: The paper is fairly written, though as discussed, many assumptions are made, and consequently further discussion of these assumptions would be beneficial.

Relation to Prior Work: The relation to prior work was discussed, however as discussed more comparison of proof techniques and discussion of other benchmarks for related problems would be beneficial. Further, more discussion of the analysis of related fixed point algorithms and results in RL would be helpful.

Reproducibility: Yes

Additional Feedback: * Line 10: I was a little confused by the last sentence, is this say that the convergence bound of the algorithm depends only logarithmically on the size of the state space? * Line 167: I believe the norm equivalence can be arbitrarily large, it just is finite for any two norms (i.e. it depends on more than a universal constant unless the constant in turn is allowed to depend on the norm). It would be beneficial to elaborate on this. * Line 210: the theorem depends on the alphas of the previous section and therefore referring to these would be beneficial.


Review 3

Summary and Contributions: The paper focused on the problem of stochastic approximation with non-smooth contraction mapping. The authors designed smooth Lyapunov function using the generalized Moreau Envelope to get finite-sample bounds. As an application, the authors proved that synchronous V-trace algorithm has finite-sample bounds. After feedback: Although I still think the proposed SA algorithm is not suitable for V-trace, the technique in SA is novel, especially using a smooth convex envelope to derive meaningful convergence rate. I believe this technique will inspire valuable ideas in designing SA algorithms. Also it is possible to derive sample complexity bound for other RL algorithms by this SA analysis. So I increase my rating from weak reject to weak accept.

Strengths: 1. The author constructed smooth relaxation for the original non-smooth Lyapunov function using Generalized Moreau Envelope. Based on this, they derived single-step contractive inequality for the relaxed Lyapunov function and obtained meaningful convergence rate. Besides, they provided a detailed analysis for the convergence rate with different stepsizes. In particular, their bounds have logarithmic dependence on the dimension. It is also notable that the noise in their setting could scale linearly as X.

Weaknesses: 1. In general, one can hardly play synchronous V-trace algorithm because the state distribution of the off-policy data can be really bad. It is also not proper to claim that the sample bounds is logarithmic in S because in each round the proposed algorithm needs at least S trajectories. 2. As far as I understand, in IMPALA, one needs to compute V_{\pi_{\bar{\rho}}} to approximate V_{\pi}. To reduce the approximation error, one should set \bar{\rho} sufficiently large. However, the sample bound in Theorem 3.1 depends on \bar{\rho} polynomially. So one can hardly say that the approximation is efficient. So I think the result about V-trace is not significant.

Correctness: Yes.

Clarity: Overall, the paper is well written and the proof is easy to follow.

Relation to Prior Work: Yes. This work considered non-smooth contraction function (e.g. L_{infty} norm) for stochastic approximation, while previous work mainly focused on the smooth ones.

Reproducibility: Yes

Additional Feedback: 1. As mentioned above, I think V-trace might not be a good application case for the proposed algorithm. That said, I hope you can provide the sample bounds for the gap between V_k and V_{\pi}. 2. In line 356-363, you mentioned how to choose \bar{c}. In general, if T is large, more samples (not trajectories) are needed. So could you provide analysis about how to minimize the number of samples?

[Author Response · NeurIPS 2020]

**R1: No experimental evaluations**: We did not focus on experiments as the goal was to demonstrate that our new technique for analyzing SA using the smoothed Lyapunov function is applicable for developing bounds for RL that can recover state-of-the-art bounds and enable new bounds for off-policy settings (please also see 'How optimal is the analysis').

**R1: The results are only valid for pseudo-contraction updates**: Our motivation for studying the SA algorithm in this setting is to analyze RL algorithms such as V-trace and $Q$-learning, which are known to have a contraction operator.

**R1: Consider only martingale-difference noise, and finite state and action spaces**: Since we consider the tabular method in RL, the underlying Markovian noise can be modeled by martingale differences. When using function approximation, Off-policy TD can potentially diverge [36]; studying it is one of our future direction.

**R1: Title is too general**: We will make the corresponding changes on the title.

**R2: A number of assumptions made**: Assumptions 2.1-2.3 in our paper are standard assumptions for studying SA algorithms involving a contraction operator, see Assumption 4.3 and Proposition 4.4 in [5]. Moreover, they are satisfied for many RL algorithms such as TD-learning and $Q$-learning, see Chapter 5 in [5]. Regarding the assumptions for the V-trace algorithm, they are from the original paper [17], and we do not make any additional assumptions.

**R2: Novelty compared to prior work**: Prior work studies either $\ell_2$-norm contraction [5,10], or contraction w.r.t. $\|\cdot\|_\infty$ under the condition that the noise is uniformly bounded by a constant [3,4]. We establish convergence rate under general norm contraction and noise whose moments scale with the current iterate. (The lack of a smooth potential function for analyzing $\|\cdot\|_\infty$-contraction SA is a long-standing open problem, and is pointed out in [5], Sec 4.3 page 154). From a technical approach, we do not decompose the analysis into one for contraction and another for noise (as has been standard in prior works [3,4]). Our joint analysis of both is the key to our recursion (Proposition 2.1).

**R2: How optimal is the analysis**: The parameters in the Generalized Moreau Envelope can be tuned to tighten the bound. Though we do not have formal results on the optimality of our bounds, our approach based on a smooth Lyapunov function recovers existing state-of-the-art finite-sample bounds for $Q$-learning that show only a logarithmic dependence on the size of the state-action space [42] in a diminishing step-size regime, and improves over [3,4] in a constant step-size regime (see Appendix I of the supplementary materials for details).

**R2: Simple ways to analyze SA with unbounded noise**: When we have $\|w_k\| \le c\|x_k\|$ (I think you are assuming that $x^* = 0$ is the fixed-point) for some small enough $c$, one can just use triangle inequality (even without taking expectation) to obtain a contractive recursion: $\|x_{k+1}\| \le (1-\epsilon_k)\|x_k\| + \epsilon_k\|\mathbf{H}(x_k)\| + \epsilon_k\|w_k\| \le (1 - (1-\gamma-c)\epsilon_k)\|x_k\|$. However, in V-trace or $Q$-learning we have *affinely* increasing noise: $\|w_k\| \le A(1 + \|x_k\|)$, and as noted in Section 3.2 and Appendix I, the coefficient $A$ is not small enough to apply this idea (and the affineness causes further difficulties).

**R2: Analyze only a particular policy evaluation algorithm**: Popular RL algorithms such TD(0), TD($n$), TD($\lambda$), $Q$-learning, and V-trace etc. can all be modeled by SA under contraction operator and martingale difference noise [5]. Thus our result is a broad tool to establish the finite-sample error bound of various RL algorithms.

**R3: Synchronous V-trace**: We agree with the reviewer that when performing asynchronous updates, there should be at least an additional factor of the dimension in the bound (indeed we see this in $Q$-learning). We will make this clearer in our paper. Studying convergence rates and concentration results for asynchronous V-trace is one of our future direction.

**R3: Regarding $V_{\pi_{\bar\rho}}$ and $V_\pi$**: When there is no clipping, we have $V_{\pi_{\bar\rho}} = V_\pi$. However, in this case the variance can be arbitrarily bad in the update, and is well recognized to be the key problem with off-policy methods. The goal of the V-trace algorithm is to reduce the variance by introducing the bias (i.e., introducing the clipper $\bar\rho$). By doing that, the variance is reduced to polynomial (quadratic) in $\bar\rho$. As for resulting bias (i.e., the gap between $V_{\pi_{\bar\rho}}$ and $V_\pi$), [17] discusses this at a high-level (Sec 4.1). A precise expression can be derived, but has complex dependencies on the behavior policy, target policy, and system parameters/dynamics. We will include this expression in the revised Supplementary material.

**R3: Polynomial dependence on $\bar\rho$**: We believe that a polynomial dependence on $\bar\rho$ is fundamental for any off-policy clipping based algorithms. Specifically, recall that if clipping is triggered, a sample is reweighted by a multiplicative factor of $\bar\rho$, which means that the signal and *noise* are both scaled by this factor. Further if this occurs for a constant fraction of time, the resulting noise variance scales order-wise as $\bar\rho^2$. Since we are looking at mean-square error, it is natural to expect a linear dependence on variance, which is what we see in our results. Thus, we believe that our results capture the correct scaling, and thus are significant for V-Trace.

**R3: Minimizing the number of samples**: For a given application, we can numerically optimize the parameters ($\bar c$, $\bar\rho$, $T$) to trade-off between contraction ratio and variance. We will discuss this in the revised draft.

[Meta-Review · NeurIPS 2020]

The reviewers did express some valid concerns and point out some points that were unclear about the paper, but overall I think the positives outweigh these issues. I would strongly encourage the authors to carefully consider all the points raised by the reviewers and update the work in the final version to address these issues as much as possible.